# CONTINUAL EVALUATION FOR LIFELONG LEARNING: IDENTIFYING THE STABILITY GAP

**Matthias De Lange, Gido M. van de Ven & Tinne Tuytelaars** *
KU Leuven

## ABSTRACT

Time-dependent data-generating distributions have proven to be difficult for gradient-based training of neural networks, as the greedy updates result in catastrophic forgetting of previously learned knowledge. Despite the progress in the field of continual learning to overcome this forgetting, we show that a set of common state-of-the-art methods still suffers from substantial forgetting upon starting to learn new tasks, except that this forgetting is temporary and followed by a phase of performance recovery. We refer to this intriguing but potentially problematic phenomenon as the *stability gap*. The stability gap had likely remained under the radar due to standard practice in the field of evaluating continual learning models only after each task. Instead, we establish a framework for *continual evaluation* that uses per-iteration evaluation and we define a new set of metrics to quantify worst-case performance. Empirically we show that experience replay, constraint-based replay, knowledge-distillation, and parameter regularization methods are all prone to the stability gap; and that the stability gap can be observed in class-, task-, and domain-incremental learning benchmarks. Additionally, a controlled experiment shows that the stability gap increases when tasks are more dissimilar. Finally, by disentangling gradients into plasticity and stability components, we propose a conceptual explanation for the stability gap.

## 1 INTRODUCTION

The fast convergence in gradient-based optimization has resulted in many successes with highly overparameterized neural networks (Krizhevsky et al., 2012; Mnih et al., 2013; Devlin et al., 2018). In the standard training paradigm, these results are conditional on having a static data-generating distribution. However, when non-stationarity is introduced by a time-varying data-generating distribution, the gradient-based updates greedily overwrite the parameters of the previous solution. This results in *catastrophic forgetting* (French, 1999) and is one of the main hurdles in *continual or lifelong learning*.

Continual learning is often presented as aspiring to learn the way humans learn, accumulating instead of substituting knowledge. To this end, many works have since focused on alleviating catastrophic forgetting with promising results, indicating such learning behavior might be tractable for artificial neural networks (De Lange et al., 2021; Parisi et al., 2019). In contrast, this work surprisingly identifies significant forgetting is still present on task transitions for standard state-of-the-art methods based on experience replay, constraint-based replay, knowledge distillation, and parameter regularization, although the observed forgetting is transient and followed by a recovery phase. We refer to this phenomenon as the **stability gap**.

**Contributions** in this work are along three main lines, with code publicly available.[1] First, we define a framework for *continual evaluation* that evaluates the learner after each update. This framework is designed to enable monitoring of the worst-case performance of continual learners from the perspective of agents that acquire knowledge over their lifetime. For this we propose novel principled metrics such as the minimum and worst-case accuracy (min-ACC and WC-ACC).

Second, we conduct an empirical study with the continual evaluation framework, which leads to identifying the stability gap, as illustrated in Figure 1, in a variety of methods and settings. An

---

*Contact at: {matthias.delange,gido.vandeven,tinne.tuytelaars}@kuleuven.be

[1]Code: https://github.com/mattdl/ContinualEvaluation

ablation study on evaluation frequency indicates continual evaluation is a necessary means to surface the stability gap, explaining why this phenomenon had remained unidentified so far. Additionally, we find that the stability gap is significantly influenced by the degree of similarity of consecutive tasks in the data stream.

Third, we propose a conceptual analysis to help explain the stability gap, by disentangling the gradients based on plasticity and stability. We do this for several methods: Experience Replay (Chaudhry et al., 2019b), GEM (Lopez-Paz & Ranzato, 2017), EWC (Kirkpatrick et al., 2017), SI (Zenke et al., 2017), and LwF (Li & Hoiem, 2017). Additional experiments with gradient analysis provide supporting evidence for the hypothesis.

**Implications of the stability gap.** (i) Continual evaluation is important, especially for safety-critical applications, as representative continual learning methods falter in maintaining robust performance during the learning process. (ii) There is a risk that sudden distribution shifts may be exploited by adversaries that can control the data stream to momentarily but substantially decrease performance. (iii) Besides these practical implications, the stability gap itself is a scientifically intriguing phenomenon that inspires further research. For example, the stability gap suggests current continual learning methods might exhibit fundamentally different learning dynamics from the human brain.

Figure 1: **The *stability gap*: substantial forgetting followed by recovery upon learning new tasks in state-of-the-art continual learning methods.** Continual evaluation at every iteration (orange curve) reveals the *stability gap*, remaining unidentified with standard task-oriented evaluation (red diamonds). Shown is the accuracy on the first task, when a network using Experience Replay sequentially learns the first five tasks of class-incremental Split-MiniImagenet. More details in Figure 2.

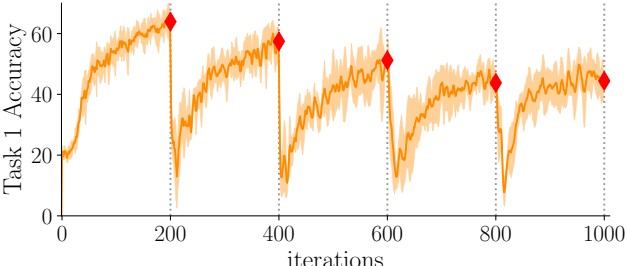

## 2 PRELIMINARIES ON CONTINUAL LEARNING

The continual or lifelong learning classification objective is to learn a function $f : \mathcal{X} \to \mathcal{Y}$ with parameters $\theta$, mapping the input space $\mathcal{X}$ to the output space $\mathcal{Y}$, from a non-stationary data stream $\mathcal{S} = \{(\mathbf{x}, \mathbf{y})_0, (\mathbf{x}, \mathbf{y})_1, ..., (\mathbf{x}, \mathbf{y})_n\}$, where data tuple $(\mathbf{x} \in \mathcal{X}, \mathbf{y} \in \mathcal{Y})_t$ is sampled from a data-generating distribution $\mathcal{D}$ which depends on time $t$. While standard machine learning assumes a static data-generating distribution, continual learning introduces the dependency on time variable $t$. This time-dependency introduces a trade-off between adaptation to the current data-generating distribution and retaining knowledge acquired from previous ones, also referred to as the *stability-plasticity trade-off* (Grossberg, 1982).

**Tasks.** The data stream is often assumed to be divided in locally stationary distributions, called tasks. We introduce discrete task identifier $k$ to indicate the $k$-th task $T_k$ with locally stationary data-generating distribution $\mathcal{D}_k$. Additionally, time variable $t$ is assumed discrete, indicating the overall iteration number in the stream, and $t_{|T_k|}$ indicates the overall iteration number at the end of task $T_k$.

**Learning continually.** During the training phase, a *learner* continuously updates $f$ based on new tuples $(\mathbf{x}, \mathbf{y})_t$ from the data stream (De Lange & Tuytelaars, 2021). Optimization follows empirical risk minimization over the observed training sets $\widetilde{D}_{\leq k}$, as the learner has no direct access to data-generating distributions $\mathcal{D}_{\leq k}$. The negative log-likelihood objective we would ideally optimize while learning task $T_k$ is:

$$\min_{\theta} \mathcal{L}_k = -\Sigma_{n=1}^{k} \mathbb{E}_{(\mathbf{x}, \mathbf{y}) \sim \widetilde{D}_n} \left[ \mathbf{y}^T \log f(\mathbf{x}; \theta) \right] \tag{1}$$

A key challenge for continual learning is to estimate this objective having only the current task's training data $\widetilde{D}_k$ available, while using limited additional compute and memory resources.

# 3   A FRAMEWORK FOR CONTINUAL EVALUATION

The *learner* continually updating $f$ has a counterpart, the continual *evaluator*, that tracks the performance of $f$ over time. Not only the data observed by the learner in data stream $\mathcal{S}$ may change over time, but also the distribution on which the learner is evaluated. This is reflected in a time dependency on the evaluator's data stream $\mathcal{S}_E$ (De Lange & Tuytelaars, 2021). The evaluator is configured by three main factors: i) the evaluation periodicity $\rho_{\text{eval}}$ determining how often $f$ is evaluated; ii) the process to construct $\mathcal{S}_E$; iii) the set of evaluation metrics.

**Evaluation periodicity.**  The typical approach to evaluation in continual learning is *task-based evaluation*: the performance of model $f$ is evaluated only after finishing training on a new task. As an alternative, here we define *continual evaluation* as evaluating the performance of $f$ every $\rho_{\text{eval}}$ training iterations. For the most fine-grained evaluation $\rho_{\text{eval}}$ is set to 1, the setting used in the following unless mentioned otherwise. The effect of increasing $\rho_{\text{eval}}$ can be observed in Figure 2 and is studied in more detail in Appendix B.

**Defining the evaluation stream.**  The desirable behavior of the model can be disentangled in performance on a set of *evaluation tasks*. We define a set of $N$ *evaluation tasks* $\mathcal{T}_E = \{E_0, ..., E_N\}$, where each evaluation task $E_i$ has a data set $\widetilde{D}_{E,i}$ sampled from data-generating distribution $\mathcal{D}_{E,i}$. The set of evaluation tasks can be extended at any point in time: $\mathcal{T}_E \leftarrow \mathcal{T}_E \cup \{E_{N+1}\}$. Following literature (van de Ven et al., 2020), we assume a new evaluation task $E_k$ is added upon each encountered training task $T_k$ in $\mathcal{S}$. Typically, the evaluation tasks are chosen to match the tasks in the training stream $\mathcal{S}$, where the evaluation data $\widetilde{D}_{E,k}$ may consist of a subset of the observed task's training data $\widetilde{D}_k$, or more commonly, consists of a held-out evaluation set to test the generalization performance with $\widetilde{D}_k \cap \widetilde{D}_{E,k} = \emptyset$. As the set of evaluation tasks expands over the training stream, we provide relaxations regarding computational feasibility in Appendix B.

**Continual evaluation metrics**  The third factor of the evaluator is the set of evaluation metrics used to assess the performance of the learner. To date, metrics have been defined mainly assuming evaluation on task transitions and focusing on the final performance of the learner. We advocate that continual worst-case performance is important for continual learners in the real world. Therefore, we propose new metrics to quantify worst-case performance (WC-ACC, min-ACC, $\text{WF}^w$) and metrics that are also applicable to task-agnostic data streams ($\text{WF}^w$, $\text{WP}^w$). We focus on classification metrics, with **accuracy** (the percentage of correctly classified instances) considered the main performance metric. Using $f_t$ to indicate the version of the model after the $t$-th overall training iteration, the accuracy of evaluation task $E_k$ at this iteration is denoted as $\mathbf{A}(E_k, f_t)$. We provide a taxonomy of metrics based on plasticity and stability in the following.

## 3.1   STABILITY-BASED METRICS

To measure stability of the learner, we aim to quantify how well knowledge from previously observed tasks $T_{<k}$ is preserved while learning new task $T_k$. **Average forgetting** (**FORG**) (Chaudhry et al., 2018) averages the accuracy difference for the most recent model $f_{t_{|T_k|}}$ compared to $f_{t_{|T_i|}}$ just after learning $T_i$ with evaluation task $E_i$, and is defined as $\frac{1}{k-1} \sum_i^{k-1} \mathbf{A}(E_i, f_{t_{|T_i|}}) - \mathbf{A}(E_i, f_{t_{|T_k|}})$. Large forgetting indicates the phenomenon of *catastrophic forgetting*, while negative forgetting indicates knowledge transfer from new to previous tasks.

For worst-case performance it is desirable to have an absolute measure on previous task performance. We define the **average minimum accuracy** (**min-ACC**) at current training task $T_k$ as the average absolute minimum accuracy over previous evaluation tasks $E_i$ after they have been learned:

$$\textbf{min-ACC}_{T_k} = \frac{1}{k-1} \sum_i^{k-1} \min_n \mathbf{A}(E_i, f_n), \ \forall t_{|T_i|} < n \le t \tag{2}$$

where the iteration number $n$ ranges from after the task is learned until current iteration $t$. This gives a worst-case measure of how well knowledge is preserved in previously observed tasks $T_{<k}$. In the following, we report the min-ACC for the last task, omitting the dependency on $T_k$ for brevity.

Furthermore, we introduce a more general stability metric that does not assume a task-based data stream, and is therefore also applicable to data incremental learning. We define **Windowed-**

**Forgetting** ($\mathbf{WF}^w$) based on a window of $w$ consecutive accuracy evaluations averaged over the evaluation set $\mathcal{T}_E$. For a single evaluation task $E_i$ the maximal accuracy decrease in the window $\Delta_{t,E_i}^{w,-}$ and the task-specific Windowed-Forgetting $\mathbf{WF}_{t,E_i}^w$ are defined at current iteration $t$ as

$$\Delta_{t,E_i}^{w,-} = \max_{m<n} \left( \mathbf{A}(E_i, f_m) - \mathbf{A}(E_i, f_n) \right), \ \forall m,n \in [t-w+1, \ t] \tag{3}$$

$$\mathbf{WF}_{t,E_i}^w = \max_n \Delta_{n,E_i}^{w,-}, \ \ \forall n \le t \tag{4}$$

Averaging the metric over all $N$ evaluation tasks results in a single metric $\mathbf{WF}_t^w = N^{-1} \sum_i^N \mathbf{WF}_{t,E_i}^w$. As it identifies the maximal observed performance drop in the window, it is considered a worst-case metric. Evaluators that can afford linear space complexity can consider the full history at iteration $t$ with $\mathrm{WF}^t$. To account for both fast forgetting and forgetting on a larger scale with constant memory, we instantiate $\mathrm{WF}^{10}$ and $\mathrm{WF}^{100}$.

## 3.2 Plasticity-based Metrics

Plasticity refers in this work to the learner's ability to acquire new knowledge from the current data-generating distribution $\mathcal{D}_k$. The **current task accuracy** measures plasticity as $\mathbf{A}(E_k, f_t)$ with $t \in \left[ \ t_{|T_{k-1}|}, \ t_{|T_k|} \ \right]$. Other metrics proposed in literature are the few-shot measure *Learning Curve Area* (Chaudhry et al., 2019a) and zero-shot *Forward Transfer* (Lopez-Paz & Ranzato, 2017). Previous metrics depend on the current learning task $T_k$ and are therefore not directly applicable to task-agnostic data streams. As the counterpart for $\mathrm{WF}^w$, we introduce the more generally applicable **Windowed Plasticity** ($\mathbf{WP}^w$), defined over all $N$ evaluation tasks by the maximal accuracy *increase* $\Delta_{t,E_i}^{w,+}$ in a window of size $w$. Appendix defines $\Delta_{t,E_i}^{w,+}$ (Eq. 10) as $\Delta_{t,E_i}^{w,-}$ but with constraint $m > n$.

$$\mathbf{WP}_t^w = \frac{1}{N} \sum_i^N \max_n \Delta_{n,E_i}^{w,+}, \ \ \forall n \le t \tag{5}$$

## 3.3 Stability-Plasticity trade-off based Metrics

The main target in continual learning is to find a balance between learning new tasks $T_k$ and retaining knowledge from previous tasks $T_{<k}$, commonly referred to as the *stability-plasticity* trade-off (Grossberg, 1982). Stability-plasticity trade-off metrics provide a single metric to quantify this balance. The standard metric for continual learning is the **Average Accuracy** (**ACC**) that after learning task $T_k$ averages performance over all evaluation tasks: $\mathbf{ACC}_{T_k} = \frac{1}{k} \sum_i^k \mathbf{A}(E_i, f_{t_{|T_k|}})$. It provides a measure of trade-off by including the accuracy of both current evaluation task $E_k$ (plasticity) and all previous evaluation tasks $E_{<k}$ (stability). The ACC is measured only at the final model $f_{t_{|T_k|}}$ and is negligent of the performance between task transitions. Therefore, we propose the **Worst-case Accuracy** (**WC-ACC**) as the trade-off between the accuracy on iteration $t$ of current task $T_k$ and the worst-case metric min-ACC$_{T_k}$ (see Eq. 2) for previous tasks:

$$\mathbf{WC\text{-}ACC}_t = \frac{1}{k} \mathbf{A}(E_k, f_t) + (1 - \frac{1}{k}) \mathbf{min\text{-}ACC}_{T_k} \tag{6}$$

This metric quantifies per iteration the minimal accuracy previous tasks retain after being learned, and the accuracy of the current task. WC-ACC gives a lower bound guarantee on ACC that is established over iterations. Evaluating after learning task $T_k$, we conclude the following lower bound:

$$\mathbf{WC\text{-}ACC}_{t_{|T_k|}} \le \mathbf{ACC}_{T_k} \tag{7}$$

## 4 Identifying the stability gap with continual evaluation

We start by conducting an empirical study confined to Experience Replay (ER) (Chaudhry et al., 2019b), as it has been shown to prevail over regularization-based methods, especially for class- and domain-incremental learning (van de Ven et al., 2022). ER stores a small subset of samples in a memory buffer, which are revisited later on by sampling mini-batches that are partly new data, partly memory data. Due to its simplicity and effectiveness against catastrophic forgetting, it has been widely adopted in literature (Rebuffi et al., 2017; Chaudhry et al., 2019b; De Lange & Tuytelaars, 2021; Chaudhry et al., 2018).

Figure 2: Average accuracy (mean±SD over 5 seeds) *on the first task* when using ER. Experiments are class-incremental (a-c) or domain-incremental (d). Evaluation periodicity ranges from standard evaluation on task transitions (red diamonds) to per-iteration continual evaluation ($\rho_{\text{eval}} = 1$). Per-iteration evaluation reveals sharp, transient drops in performance when learning new tasks: the *stability gap*. Vertical lines indicate task transitions; horizontal lines the min-ACC averaged over seeds.

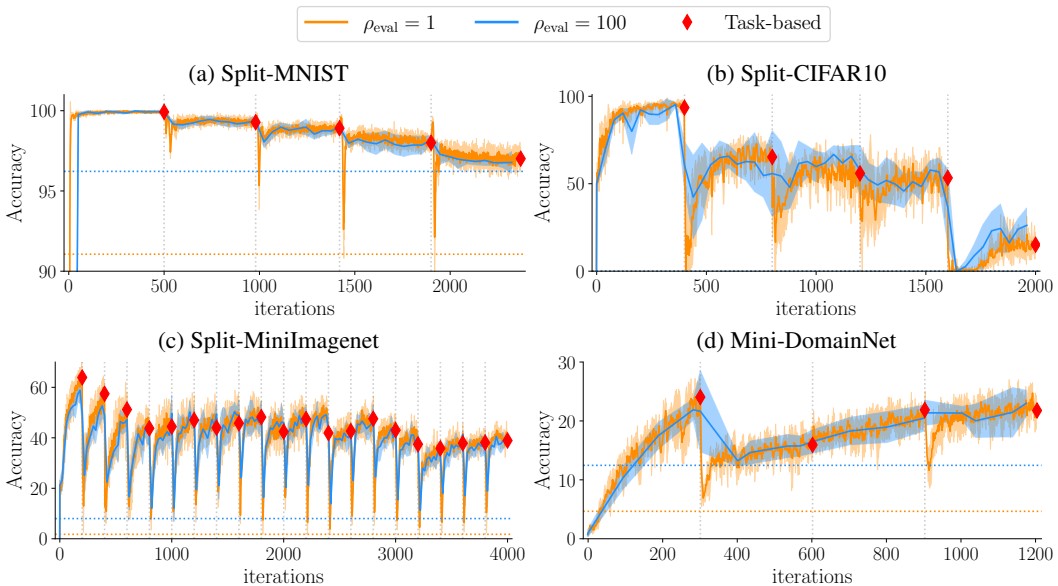

**Datasets.** For experiments on *class-incremental learning* we use three standard datasets: MNIST (LeCun & Cortes, 2010) consists of grayscale handwritten digits, CIFAR10 (Krizhevsky et al., 2009) contains images from a range of vehicles and animals, and MiniImagenet (Vinyals et al., 2016) is a subset of Imagenet (Russakovsky et al., 2015). **Split-MNIST**, **Split-CIFAR10**, and **Split-MiniImagenet** are defined by respectively splitting the data in 5, 5, and 20 tasks based on the 10, 10, and 100 classes. For *domain-incremental learning* we consider drastic domain changes in **Mini-DomainNet** (Zhou et al., 2021), a scaled-down subset of 126 classes of DomainNet (Peng et al., 2019) with over 90k images, considering domains: clipart, painting, real, sketch.

**Setup.** We employ continual evaluation with evaluation periodicity in range $\rho_{\text{eval}} \in \{1, 10, 10^2, 10^3\}$ and subset size 1k per evaluation task, based on our feasibility analysis in Appendix B. For reference with literature, task-transition based metrics ACC and FORG follow standard evaluation with the entire test set. Split-MNIST uses an MLP with 2 hidden layers of 400 units. Split-CIFAR10, Split-MiniImagenet and Mini-DomainNet use a slim version of Resnet18 (Lopez-Paz & Ranzato, 2017). SGD optimization is used with 0.9 momentum. To make sure our worst-case analysis applies to the best-case configuration for ER, we run a gridsearch over different hyperparameters and select the entry with the highest stability-plasticity trade-off metric ACC on the held-out evaluation data (Lopez-Paz & Ranzato, 2017). Details for all experiments can be found in Appendix C and code is available here: https://github.com/mattdl/ContinualEvaluation.

**Qualitative analysis with continual evaluation.** Figure 2 illustrates the first task accuracy curves for ER over the 4 benchmarks. The red markers on task transitions indicate the standard evaluation scheme in continual learning. We find that continual evaluation ($\rho_{\text{eval}} = 1$) reveals significant temporary forgetting between task transitions, both for the class- and domain-incremental benchmarks. After the significant performance drops, partial recovery follows, making the task-transition based metrics such as ACC and FORG bad worst-case performance estimators. We refer to this *phenomenon of transient, significant forgetting when learning new tasks* as the **stability gap**.

**Quantifying worst-case performance.** The qualitative analysis allows observation of the stability gap from the accuracy curves over time. However, performance analysis also requires a quantifying measure for the worst-case performance. For class-incremental Split-MiniImagenet, we report results

for our new continual evaluation metrics in Table 1, from which two important observations can be made. First, we confirm that the standard metric ACC measured on task transitions is negligent to the stability gap, whereas our worst-case performance metrics min-ACC and WC-ACC clearly indicate the sheer performance loss. Second, our metrics indicate similarly to Figure 2 that fine-grained evaluation periodicity is crucial to identify the stability gap. Table 1 shows that the sharp performance drops are phased out as min-ACC and WC-ACC both increase with evaluation periodicity $\rho_{\text{eval}}$, and large-scale forgetting $WF^{100}$ decreases. Table 3 in Appendix reports similar results on the other benchmarks. Additionally, Appendix D reports results for the stability gap for various ER buffer sizes, in a speech modality experiment, and for *online* continual learning.

Table 1: Our newly proposed continual evaluation metrics on class-incremental Split-MiniImagenet for a range of evaluation periodicities $\rho_{\text{eval}}$. Standard continual learning metrics are $32.9 \pm 0.8$ (ACC) and $32.3 \pm 1.0$ (FORG). Results over 5 seeds reported as mean ($\pm$SD).

| | Trade-off | Stability | | | Plasticity | |
|---|---|---|---|---|---|---|
| $\rho_{\text{eval}}$ | WC-ACC | min-ACC | $WF^{10}$ | $WF^{100}$ | $WP^{10}$ | $WP^{100}$ |
| $10^0$ | $4.1 \pm 0.3$ | $0.5 \pm 0.2$ | $56.6 \pm 0.9$ | $64.6 \pm 1.1$ | $49.3 \pm 1.6$ | $67.6 \pm 0.7$ |
| $10^1$ | $5.0 \pm 0.4$ | $1.4 \pm 0.5$ | $60.9 \pm 0.5$ | $63.0 \pm 0.6$ | $65.3 \pm 0.3$ | $68.1 \pm 0.4$ |
| $10^2$ | $6.7 \pm 0.4$ | $3.1 \pm 0.4$ | $58.8 \pm 0.6$ | $60.5 \pm 0.7$ | $67.0 \pm 0.7$ | $67.2 \pm 0.8$ |
| $10^3$ | $7.1 \pm 1.1$ | $3.6 \pm 1.1$ | $57.7 \pm 0.5$ | $59.3 \pm 0.3$ | $66.1 \pm 0.8$ | $66.3 \pm 0.9$ |

## 4.1 THE INFLUENCE OF TASK SIMILARITY ON THE STABILITY GAP

When the domain shift between subsequent tasks in the stream increases, the interference of the objectives is expected to lead to higher forgetting (FORG) and hence lower average accuracy (ACC) over all tasks. However, the effect on the stability gap remains unclear.

**Experiment setup.** We set up a controlled domain-incremental learning experiment with Rotated-MNIST, where each task constitutes the entire MNIST dataset with an increasing but fixed rotation transform of $\phi$ degrees. The increasing task-specific rotation results in controllable domain shifts over tasks in the input space. To avoid ambiguity between digits 6 and 9, we constrain the total rotation to not exceed $180°$. We consider a cumulative fixed rotation by $\phi$ degrees, leading to $\phi_0$, $(\phi_0 + \phi)$, and $(\phi_0 + 2\phi)$ for the three tasks, with $\phi_0$ set to $-90°$ for the initial task's rotation. To increase the task dissimilarity, we consider a range of increasing relative rotations $\phi = [10°, 30°, 60°, 80°]$. We study ER with a buffer capacity of 1k samples.

**Results.** Table 2 confirms the decrease in ACC as the distribution shift between tasks increases. Between the easiest ($\phi = 10$) and hardest ($\phi = 80$) sequence, ACC declines with $6.6\%$. However, the effect on the stability gap is substantially larger as the min-ACC drops from $94.3\%$ to $69.2\%$, a $25.1\%$ decrease. Similarly, FORG between $\phi = 60$ and $\phi = 80$ indicates only $1.1\%$ more forgetting, whereas the $WF^{10}$ shows a decline of $5.0\%$. This indicates both that i) the standard metrics fail to capture the effects of the stability gap, and ii) the stability gap increases significantly for larger distribution shifts. As additional confirmation, the accuracy curves for the first task in Figure 3 show qualitatively that the stability gap increases substantially with increasing rotations.

Table 2: Task-similarity results for ER in Rotated-MNIST. Three subsequent tasks are constructed by rotating MNIST images by $\phi$ degrees per task. We decrease task similarity by increasing the rotation from $\phi = 10°$ to $\phi = 80°$. Results over 5 seeds reported as mean ($\pm$SD).

| | Trade-off | | Stability | | | | Plasticity | |
|---|---|---|---|---|---|---|---|---|
| $\phi$ | ACC | WC-ACC | min-ACC | FORG | $WF^{10}$ | $WF^{100}$ | $WP^{10}$ | $WP^{100}$ |
| $10°$ | $96.6 \pm 0.3$ | $95.3 \pm 0.3$ | $94.3 \pm 0.3$ | $0.7 \pm 0.3$ | $3.5 \pm 0.3$ | $3.6 \pm 0.2$ | $20.9 \pm 0.9$ | $30.7 \pm 1.5$ |
| $30°$ | $93.7 \pm 0.5$ | $91.8 \pm 0.4$ | $89.3 \pm 0.6$ | $4.6 \pm 0.6$ | $5.1 \pm 0.7$ | $6.0 \pm 0.4$ | $28.2 \pm 1.1$ | $41.3 \pm 1.6$ |
| $60°$ | $90.6 \pm 0.6$ | $83.5 \pm 0.8$ | $77.3 \pm 1.5$ | $8.7 \pm 0.8$ | $12.8 \pm 1.4$ | $13.5 \pm 1.2$ | $50.8 \pm 1.0$ | $67.2 \pm 0.9$ |
| $80°$ | $90.0 \pm 0.5$ | $78.4 \pm 1.4$ | $69.2 \pm 2.2$ | $9.8 \pm 0.6$ | $17.8 \pm 2.0$ | $18.6 \pm 1.7$ | $57.3 \pm 1.7$ | $75.6 \pm 1.8$ |

Figure 3: Controlled task similarity experiment with task rotation angle $\phi$ in Rotated-MNIST. Accuracy curves for ER in the first task are reported as mean ($\pm$SD) over 5 seeds. The two vertical lines indicate task transitions. Horizontal lines indicate the min-ACC averaged over seeds.

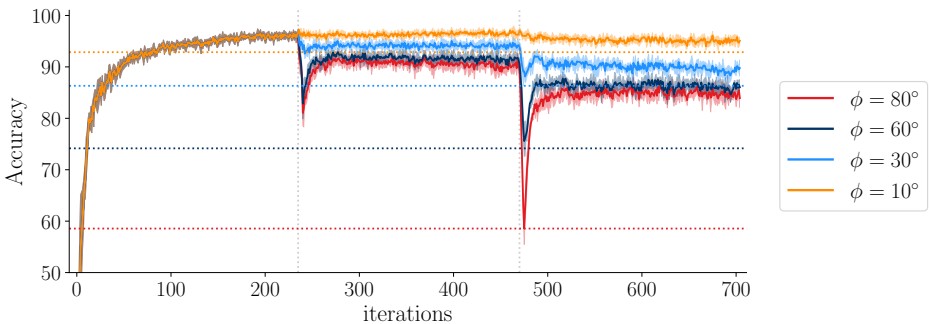

## 5 CONCEPTUAL ANALYSIS OF THE STABILITY GAP

To establish a conceptual grounding for the stability gap phenomenon, we disentangle the continual learning gradients of the objective $\mathcal{L}$ in $\alpha$-weighed plasticity and stability gradients

$$\nabla\mathcal{L} = \alpha\nabla\mathcal{L}_{\text{plasticity}} + (1 - \alpha)\nabla\mathcal{L}_{\text{stability}} \tag{8}$$

In gradient-based optimization of model $f$, $\nabla\mathcal{L}_{\text{plasticity}}$ aims to improve performance on the current task and $\nabla\mathcal{L}_{\text{stability}}$ maintains performance on past tasks. In the viewpoint of the stability-plasticity trade-off (Grossberg, 1982), continual learning methods aim to balance both gradient terms. However, due to task-oriented evaluation and lack of continual evaluation, progress in this balance has only been verified on the task transitions.

**Finetuning** optimizes solely for the current task $T_k$ and is not concerned with stability, with $||\nabla\mathcal{L}_{\text{stability}}|| = 0$. The lack of a stability term results in greedy updates for the currently observed data, resulting in forgetting on distribution shifts. Our continual evaluation framework indicates that severe forgetting of past tasks occurs already during the first few training iterations on a new task (see Appendix D.3).

**Experience replay (ER)** learns concurrently from data of the new task and a subset of previous task data sampled from experience buffer $\mathcal{M}$ of size $M$. The loss term $\mathcal{L}_{\text{stability}}$ is obtained by revisiting previous task samples in $\mathcal{M}$. In contrast to finetuning, this results in a stability gradient $\nabla\mathcal{L}_{\text{stability}}$ designed to prevent catastrophic forgetting.

**Forgetting.** Our further analysis of the $\nabla\mathcal{L}_{\text{stability}}$ gradient dynamics indicate a low gradient norm during the early phase of training on a new task. We first consider a data stream with two subsequent tasks. When training starts on $T_2$, $f$ presumably has converged for the first task $T_1$, resulting in $||\nabla\mathcal{L}_{T_1}|| \approx 0$. Therefore, directly after the task transition, we indeed have $||\nabla\mathcal{L}_{\text{stability}}|| \approx 0$ because the replayed samples are exclusively from $T_1$. Generalizing this to longer task sequences requires not only nearly zero gradients for the previous task, but for all previous task data in the entire replay buffer $\mathcal{M}$. This has been empirically confirmed by Verwimp et al. (2021) indicating ER to consistently converge to near-zero gradients for $\mathcal{M}$. We demonstrate similar findings for Split-MNIST in Figure 4(e-h). Due to the imbalance of the stability and plasticity gradients, the plasticity gradient will dominate and potentially result in forgetting. Our empirical findings indeed confirm for ER that significant drops in accuracy occur directly after the task transition, as shown in Figure 2.

**Recovery.** Upon the initial steps of learning a new task with the greedy $\nabla\mathcal{L}_{\text{plasticity}}$ updates, parameters change with nearly no gradient signal from replay buffer $\mathcal{M}$. These updates in turn lead to increasing gradient norm of $\nabla\mathcal{L}_{\text{stability}}$, allowing the samples in $\mathcal{M}$ to be re-learned.

*In contrast to the prior belief of $\nabla\mathcal{L}_{\text{stability}}$ maintaining prior knowledge, we find that stability is at least partially preserved by means of relearning, leading to the recovery of prior knowledge.* This is confirmed by our experiments in Section 4 on five class- and domain-incremental benchmarks.

## 5.1 FINDING A STABILITY GAP IN OTHER CONTINUAL LEARNING METHODS

Our analysis of disentangling the gradients based on plasticity and stability also applies to other representative methods in continual learning. Additionally, we provide empirical evidence for the stability gap in these methods, focusing on constraint-based replay method GEM (Lopez-Paz & Ranzato, 2017), knowledge-distillation method LwF (Li & Hoiem, 2017), and parameter regularization methods EWC (Kirkpatrick et al., 2017) and SI (Zenke et al., 2017). The details of the experiments are in Appendix C.

Figure 4: GEM and ER accuracy curves (*a-d*) on class-incremental Split-MNIST for the first four tasks, and the per-iteration L2-norms of current $\nabla\mathcal{L}_{\text{stability}}$ (*e-h*). Results reported as mean ($\pm$SD) over 5 seeds, with horizontal lines representing average min-ACC. Vertical lines indicate the start of a new task. Note that the x-axis scale varies over (*a-d*) and (*e-h*) are zooms of the first 50 iterations.

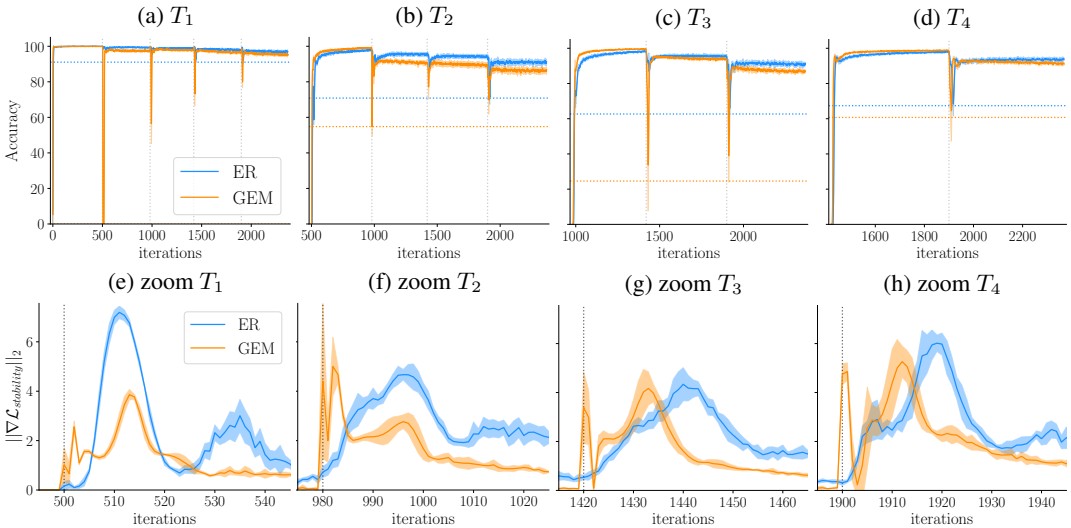

**Gradient-constrained replay.** Gradient Episodic memory (GEM) (Lopez-Paz & Ranzato, 2017) exploits a memory buffer $\mathcal{M}$ similar to ER, divided over $K$ equally sized task buffers $\mathcal{M}_k$. However, instead of directly optimizing the objective for the samples in $\mathcal{M}$, their task-specific gradients $g_k = \nabla\mathcal{L}(\mathcal{M}_k)$ are used to compose a set of constraints. The constraints $\langle g_t, g_n \rangle \geq 0, \ \forall n < k$ attempt to prevent loss increase on the $k-1$ previous tasks, with $g_t = \nabla\mathcal{L}_{\text{plasticity}}$ the gradient of the current observed sample $(\mathbf{x},\mathbf{y})_t$ in task $T_k$. Current gradient $g_t$ is projected to the closest gradient $\tilde{g}$ satisfying the constraint set, obtained by Quadratic Programming. We reformulate the projection to obtain the conditional stability gradient:

$$\nabla\mathcal{L}_{\text{stability}} = \begin{cases} \vec{0}, & \text{if } \langle g_t, g_n \rangle \geq 0, \ \forall n < k \\ \tilde{g} - g_t, & \text{otherwise} \end{cases} \tag{9}$$

As the GEM constraints explicitly attempt to prevent an increase in previous task losses, $||\nabla\mathcal{L}_{\text{stability}}||$ is only zero if the current gradient update $g_t$ lies within the feasible region of the constraint set or is near-zero with only slightly violated constraints $||\tilde{g} - g_t|| \approx 0$. As in the dot-product constraints the sign is determined solely based on the gradient angles, satisfying them is independent of the norm of $||\nabla\mathcal{L}_{\text{stability}}||$ on task transitions. This suggests that GEM might enable avoiding or alleviating the stability gap.

However, empirically we find that also GEM is prone to the stability gap (Figure 4). Compared to ER, the stability gap of GEM is significantly larger, indicated by the large discrepancy in the horizontal lines representing average min-ACC. On the task transitions for Split-MNIST, Figure 4(*e-h*) shows that GEM has significant $||\nabla\mathcal{L}_{\text{stability}}||$ compared to ER (determined following Eq. 9), indicating large violations of the constraints. However, especially for task transitions $T_3$ and $T_4$ we observe the gradients to drop to near-zero magnitude, resulting in a few updates mostly based on $\nabla\mathcal{L}_{\text{plasticity}}$.

**Distillation-based methods** (Li & Hoiem, 2017; Rannen et al., 2017) prevent forgetting of a previous task $T_{k-1}$ by penalizing changes in the output distribution for samples in the current task $\mathbf{x} \in \tilde{D}_k$.

This results in a regularization objective $\mathcal{L}_{\text{stability}} = \text{KL}(f_{t_{|T_{k-1}|}}(\mathbf{x}_t)||f_t(\mathbf{x}_t))$ that distills knowledge via KL-divergence (Hinton et al., 2015) from the previous task model at $t_{|T_{k-1}|}$ to the current model. Before the first update on a new task, the previous and current task models are identical, leading to $||\nabla\mathcal{L}_{\text{stability}}|| = 0$ caused by perfectly matching distributions, which could lead to a stability gap.

**Parameter regularization methods** (Kirkpatrick et al., 2017; Zenke et al., 2017; Aljundi et al., 2018; Kao et al., 2021), also referred to as model-prior based methods, penalize changes in important parameters for previous tasks with regularization objective $\mathcal{L}_{\text{stability}} = (\theta - \theta^*)^T \mathbf{\Omega} (\theta - \theta^*)$, where parameter importance is defined by the penalty matrix $\mathbf{\Omega} \in \mathbb{R}^{|\theta|\times|\theta|}$ and weighed by the change of the model's parameters $\theta$ w.r.t. the previous task solution $\theta^* = \theta_{t_{|T_{k-1}|}}$. For the first updates on a new task, the parameters $\theta$ are still close to the previous task solution $\theta^*$ resulting in $||\nabla\mathcal{L}_{\text{stability}}|| \approx 0$, and are even identical for the first update, resulting in $||\nabla\mathcal{L}_{\text{stability}}|| = 0$.

Figure 5 empirically confirms the stability gap for representative distillation-based (LwF) and parameter regularization methods (EWC, SI) in two settings: first, task-incremental Split-MNIST, where each task is allocated a separate classifier (Figure 5(a-d)); second, Rotated-MNIST from Section 4.1 with lowest task similarity ($\phi = 80°$) between the 3 tasks. Setup details are reported in Appendix C.

Figure 5: Distillation-based (LwF) and parameter regularization methods (EWC, SI) accuracy curves on the first four tasks (*a-d*) of task-incremental Split-MNIST, with a separate classifier head per task. The same methods are considered for domain-incremental Rotated-MNIST ($\phi = 80°$) for the first two tasks (*e-f*) out of three. Results reported as mean ($\pm$SD) over 5 seeds, with horizontal lines representing average min-ACC. Vertical lines indicate the start of a new task.

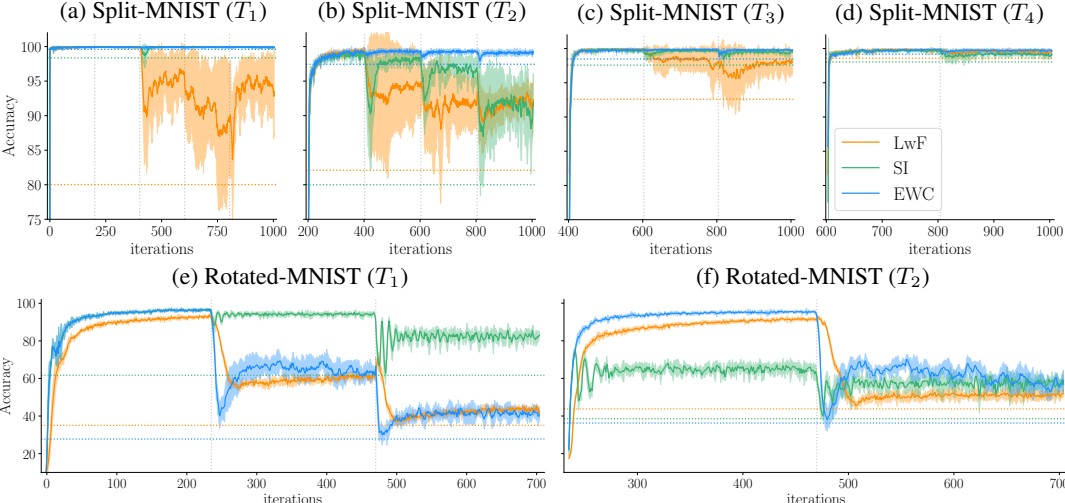

## 6  CONCLUSION

This work proposed a novel framework for *continual evaluation* with new metrics that enable measuring worst-case performance and are applicable to task-agnostic data streams. Our evaluation framework identified shortcomings of the standard task-oriented evaluation protocol for continual learning, as we identified a striking and potentially problematic phenomenon: the stability gap. In our study on seven continual learning benchmarks, we showed that upon starting to learn a new task, various state-of-the-art continual learning methods suffer from a significant loss in performance on previously learned tasks that, intriguingly, is often recovered later on. We found the stability gap increasing significantly as subsequent tasks are more dissimilar. To provide insight into what might be causing the stability gap, we formalized a conceptual analysis of the gradients, disentangled into plasticity and stability terms. This led to consistent findings for experience replay, knowledge-distillation, and parameter regularization methods, but left some open questions for constraint-based replay. Interesting directions for future work include mechanisms to overcome the stability gap and connections to biological neural networks: when learning something new, does the brain suffer from transient forgetting as well?

ACKNOWLEDGMENTS

This project has received funding from the ERC project KeepOnLearning (reference number 101021347), the KU Leuven C1 project Macchina, and the European Union's Horizon 2020 research and innovation program under the Marie Skłodowska-Curie grant agreement No 101067759.

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

SUPPLEMENTAL MATERIAL

Supplemental materials include a discussion on the limitations and societal impact of this work (Appendix A), an empirical study and discussion on computational feasibility for continual evaluation (Appendix B), the detailed reproducibility details for all experiments (Appendix C), and additional empirical evidence in Appendix D.

## A  LIMITATIONS AND SOCIETAL IMPACT

**Computational complexity of continual evaluation.**  Both time and space complexity are important factors for continual learning with constrained resources. The space complexity for continual evaluation retains a linear increase with the number of tasks $k$ as for standard evaluation. However, the time complexity changes from a task-based periodicity to a per-iteration periodicity. The ratio of computational increase for a single task $T_k$ can be defined as $\frac{t_{|T_k|} - t_{|T_{k-1}|}}{\rho_{\text{eval}}}$ where $t_{|T_k|} - t_{|T_{k-1}|}$ is the number of iterations for that task. Albeit the increase in time complexity, due to our empirical findings, we advocate the use of continual evaluation when possible for analysis of the learning behavior and especially for safety-critical applications. We refer to Appendix B to enable tractable continual evaluation.

**Parameter isolation methods** (De Lange et al., 2021) are not considered in our empirical study for the stability gap, as typically fixed parts of the model are allocated to specific tasks (Rusu et al., 2016; Verma et al., 2021; Serra et al., 2018; Mallya & Lazebnik, 2018). The task identifier is typically assumed available for inference, allowing to activate only the subnetwork for that task. This allows to prevent any changes and hence forgetting for observed tasks, albeit impeding backward transfer. Therefore, for methods where no changes in performance of learned tasks is allowed, a stability gap cannot be present either.

**Societal impact.**  A continual learning agent operational in the real world may lead to unpredictable results as the agent may observe malicious data points, learn from highly biased data, or learn undesirable behavior that diverges from its original goals. Such evolution can remain undetected with sparse or even no evaluation of the learner. Additionally, the sudden distribution shifts may be exploited by adversaries to momentarily but significantly decrease performance. Therefore, we deem continual evaluation important for fine-grained monitoring of the agent's learning behavior.

## B  TRACTABLE CONTINUAL EVALUATION

The challenge in continual evaluation is twofold. First, the high evaluation frequency is computationally demanding. Second, the evaluator's set of locally stationary test distributions may expand with the number of tasks. Furthermore, we discuss practical considerations for real-world applications by using training data for evaluation and parallelism.

**Evaluation periodicity.**  A first relaxation for feasible continual evaluation is to increase the per-update evaluation periodicity $\rho_{\text{eval}} = 1$ to larger values. We perform an analysis for $\rho_{\text{eval}} \in \{1, 10, 10^2, 10^3\}$ on 4 benchmarks for Experience Replay (ER). We follow the full setup and datasets discussed in Section 4 and Appendix C. Important for this analysis are our findings in Section 4 showing that continual evaluation reveals catastrophic drops in performance after task transitions. Table 1 shows for Split-MiniImagenet that these sharp drops are phased out as min-ACC and WC-ACC both increase with evaluation periodicity $\rho_{\text{eval}}$, and large-scale forgetting $\text{WF}^{100}$ decreases. This phenomenon is illustrated in Figure 2, where further increasing periodicity up to the standard evaluation on task transitions becomes entirely neglicent of these performance drops. Therefore, continual evaluation of our experiments in the main paper adopt $\rho_{\text{eval}} = 1$ unless mentioned otherwise.

Additionally to the Split-MiniImagenet ER results, we additionally report the results for Split-MNIST, Split-CIFAR10, and Mini-DomainNet in Table 3. We find consistent conclusions as for Split-MiniImagenet, i.e. the stability gap is unobserved for larger evaluation periodicity, especially notable for $\rho_{\text{eval}} \in \{100, 1000\}$ which are indicated in bold. The subsample size of the evaluation sets is 1000 as for the other experiments in this work.

We noticed that results in Split-CIFAR10 have large variance in min-ACC (and hence WC-ACC) for $\rho_{\text{eval}} = 100$. As the Split-CIFAR10 accuracy is noisy over the iterations as indicated in Figure 2, we found in some runs for $\rho_{\text{eval}} = 100$ to completely miss the accuracy drop to zero of $T_4$ for 2 out of the 5 seeds. This dependency on initialization seed stresses the importance of smaller $\rho_{\text{eval}}$ to reduce dependency on the exact point of evaluation.

Table 3: Full results on Split-MNIST, Split-CIFAR10, Split-MiniImagenet and Mini-DomainNet in continual evaluation metrics for a range of evaluation periodicities $\rho_{\text{eval}}$. Results over 5 seeds reported as mean ($\pm$SD). The Split-MiniImagenet subset of the full results is reported in the main paper in Table 1.

| $\rho_{\text{eval}}$ | Trade-off | | Stability | | Plasticity | |
|---|---|---|---|---|---|---|
| | WC-ACC | min-ACC | WF$^{10}$ | WF$^{100}$ | WP$^{10}$ | WP$^{100}$ |
| **Split-MNIST** | | | | | | |
| 1 | $77.7_{\pm 3.5}$ | $73.0_{\pm 4.4}$ | $18.4_{\pm 3.7}$ | $21.2_{\pm 3.7}$ | $84.9_{\pm 1.5}$ | $95.5_{\pm 1.5}$ |
| 10 | $81.3_{\pm 0.9}$ | $77.4_{\pm 1.1}$ | $16.4_{\pm 1.1}$ | $17.2_{\pm 1.1}$ | $94.5_{\pm 1.2}$ | $97.5_{\pm 1.4}$ |
| $10^2$ | $92.3_{\pm 0.2}$ | $\mathbf{91.3_{\pm 0.2}}$ | $4.6_{\pm 0.3}$ | $\mathbf{6.0_{\pm 0.3}}$ | $97.1_{\pm 1.4}$ | $97.2_{\pm 1.5}$ |
| $10^3$ | $92.3_{\pm 0.2}$ | $\mathbf{91.3_{\pm 0.2}}$ | $4.6_{\pm 0.3}$ | $\mathbf{6.0_{\pm 0.3}}$ | $97.1_{\pm 1.4}$ | $97.2_{\pm 1.5}$ |
| **Split-CIFAR10** | | | | | | |
| 1 | $17.9_{\pm 0.9}$ | $0.0_{\pm 0.0}$ | $71.1_{\pm 1.8}$ | $76.0_{\pm 2.0}$ | $57.7_{\pm 2.3}$ | $81.4_{\pm 1.1}$ |
| 10 | $18.3_{\pm 0.5}$ | $0.1_{\pm 0.2}$ | $71.0_{\pm 3.3}$ | $74.4_{\pm 1.7}$ | $78.0_{\pm 3.3}$ | $89.2_{\pm 0.9}$ |
| $10^2$ | $17.4_{\pm 1.8}$ | $\mathbf{2.0_{\pm 2.5}}$ | $65.9_{\pm 3.1}$ | $\mathbf{71.4_{\pm 3.2}}$ | $86.9_{\pm 1.8}$ | $87.1_{\pm 1.6}$ |
| $10^3$ | $17.0_{\pm 1.1}$ | $\mathbf{0.4_{\pm 0.5}}$ | $66.0_{\pm 3.4}$ | $\mathbf{70.2_{\pm 1.7}}$ | $86.0_{\pm 1.9}$ | $86.1_{\pm 1.9}$ |
| **Split-MiniImagenet** | | | | | | |
| 1 | $4.1_{\pm 0.3}$ | $0.5_{\pm 0.2}$ | $56.6_{\pm 0.9}$ | $64.6_{\pm 1.1}$ | $49.3_{\pm 1.6}$ | $67.6_{\pm 0.7}$ |
| 10 | $5.0_{\pm 0.4}$ | $1.4_{\pm 0.5}$ | $60.9_{\pm 0.5}$ | $63.0_{\pm 0.6}$ | $65.3_{\pm 0.3}$ | $68.1_{\pm 0.4}$ |
| $10^2$ | $6.7_{\pm 0.4}$ | $\mathbf{3.1_{\pm 0.4}}$ | $58.8_{\pm 0.6}$ | $\mathbf{60.5_{\pm 0.7}}$ | $67.0_{\pm 0.7}$ | $67.2_{\pm 0.8}$ |
| $10^3$ | $7.1_{\pm 1.1}$ | $\mathbf{3.6_{\pm 1.1}}$ | $57.7_{\pm 0.5}$ | $\mathbf{59.3_{\pm 0.3}}$ | $66.1_{\pm 0.8}$ | $66.3_{\pm 0.9}$ |
| **Mini-DomainNet** | | | | | | |
| 1 | $13.7_{\pm 1.0}$ | $9.6_{\pm 1.0}$ | $15.1_{\pm 1.4}$ | $17.8_{\pm 1.2}$ | $10.2_{\pm 0.6}$ | $16.9_{\pm 0.7}$ |
| 10 | $15.1_{\pm 1.6}$ | $11.0_{\pm 1.7}$ | $14.8_{\pm 0.5}$ | $15.7_{\pm 0.5}$ | $14.3_{\pm 0.9}$ | $22.0_{\pm 0.9}$ |
| $10^2$ | $18.3_{\pm 1.1}$ | $\mathbf{15.8_{\pm 1.3}}$ | $10.0_{\pm 0.8}$ | $\mathbf{10.0_{\pm 0.8}}$ | $20.6_{\pm 1.0}$ | $21.0_{\pm 0.8}$ |
| $10^3$ | $18.9_{\pm 1.8}$ | $\mathbf{16.5_{\pm 1.9}}$ | $10.0_{\pm 1.2}$ | $\mathbf{10.0_{\pm 1.2}}$ | $21.0_{\pm 1.6}$ | $21.0_{\pm 1.6}$ |

**Subsampling the evaluation sets.** A second relaxation for feasible continual evaluation is to control the number of evaluation samples. The set of evaluation tasks $\mathcal{T}_E$ often grows linearly over time. This is unavoidable if we want to provide data-based performance guarantees per learned task. Nonetheless, we can reduce the absolute computation time by limiting the number of samples per evaluation set. With $\rho_{\text{eval}} = 1$ and uniformly subsampling each evaluation task's dataset $\tilde{D}_{E,i}$, Table 4 indicates for Split-MNIST, Split-CIFAR10, and Split-MiniImagenet that a sample size of 1k provides a good approximation for using the entire test set ('*All*'). A smaller sample size of 100 induces more noise in the performance estimate, resulting in larger deviations compared to the entire dataset, especially notable for WF$^{100}$ and WP$^{100}$.

Instead of the uniform sampling, an alternative but more complex sampling scheme would also be possible to restrain the number of evaluation samples. For example by measuring task similarity (e.g. via task2vec (Achille et al., 2019)) and merging test data of similar tasks in fixed capacity bins. Although this would be applicable for real-world learners, in our analysis we opt for the straightforward subsampling to avoid confounding factors rooted in the task similarity procedure.

**Exploiting training data.** A disadvantage of extracting held-out evaluation data from the data stream is that this data cannot be used by the learner. Focusing on Split-MiniImagenet, Table 4 shows the continual evaluation of ER with training subsets compared to the actual evaluation data, with similar ACC of $31.5 \pm 0.5$ for the train set and even slightly higher $32.7 \pm 1.3$ for the test set. We find that worst-case stability metric min-ACC is very similar on both sets, and catastrophic forgetting in WF$^{100}$ is significantly higher for the training data. This suggests it can serve as a good (over)estimator to monitor the stability of the learner.

**Parallellism.** In real-world continual evaluation, parallelism can be exploited by separating the *learner* and *evaluator* in different processing units. The evaluator captures a snapshot from the model

Table 4: Continual evaluation metrics for different sample sizes of the evaluation sets and comparison to training set performance on 3 benchmarks with $\rho_{\text{eval}} = 1$. '*All*' indicates the full task test sets. Results over 5 seeds reported as mean ($\pm$SD). The $\mathbf{10^3}$ in bold indicates the setting we applied for the other experiments in this work.

| | | Trade-off | | Stability | | Plasticity | |
|---|---|---|---|---|---|---|---|
| | Sample size | **WC-ACC** | **min-ACC** | $\mathbf{WF^{10}}$ | $\mathbf{WF^{100}}$ | $\mathbf{WP^{10}}$ | $\mathbf{WP^{100}}$ |
| **Split-MNIST** | | | | | | | |
| Eval | All | $77.2 \pm 1.4$ | $72.3 \pm 1.8$ | $18.2 \pm 1.6$ | $20.8 \pm 1.3$ | $85.3 \pm 1.3$ | $95.0 \pm 1.4$ |
| | $\mathbf{10^3}$ | $77.7 \pm 3.5$ | $73.0 \pm 4.4$ | $18.4 \pm 3.7$ | $21.2 \pm 3.7$ | $84.9 \pm 1.5$ | $95.5 \pm 1.5$ |
| | $10^2$ | $75.1 \pm 2.8$ | $69.4 \pm 3.6$ | $25.0 \pm 3.4$ | $27.8 \pm 3.3$ | $86.9 \pm 1.6$ | $98.0 \pm 1.2$ |
| Train | $10^3$ | $77.6 \pm 3.5$ | $72.6 \pm 4.5$ | $19.2 \pm 4.0$ | $22.0 \pm 3.8$ | $85.2 \pm 1.8$ | $95.7 \pm 1.4$ |
| **Split-CIFAR10** | | | | | | | |
| Eval | All | $18.1 \pm 0.3$ | $0.0 \pm 0.0$ | $73.0 \pm 5.4$ | $77.8 \pm 4.9$ | $58.6 \pm 3.0$ | $81.5 \pm 4.8$ |
| | $\mathbf{10^3}$ | $17.8 \pm 0.6$ | $0.0 \pm 0.0$ | $71.3 \pm 4.2$ | $75.7 \pm 2.6$ | $59.2 \pm 2.7$ | $82.0 \pm 3.3$ |
| | $10^2$ | $18.3 \pm 0.6$ | $0.0 \pm 0.0$ | $73.9 \pm 4.3$ | $81.7 \pm 2.3$ | $59.4 \pm 2.5$ | $86.8 \pm 1.7$ |
| Train | $10^3$ | $19.3 \pm 0.9$ | $0.0 \pm 0.0$ | $75.8 \pm 5.3$ | $79.8 \pm 2.7$ | $57.8 \pm 2.8$ | $81.8 \pm 2.5$ |
| **Split-MiniImagenet** | | | | | | | |
| Eval | All | $4.1 \pm 0.3$ | $0.5 \pm 0.2$ | $56.0 \pm 0.7$ | $64.2 \pm 0.6$ | $48.3 \pm 1.8$ | $67.8 \pm 0.4$ |
| | $\mathbf{10^3}$ | $4.1 \pm 0.3$ | $0.5 \pm 0.2$ | $56.6 \pm 0.9$ | $64.6 \pm 1.1$ | $49.3 \pm 1.6$ | $67.6 \pm 0.7$ |
| | $10^2$ | $3.9 \pm 0.5$ | $0.3 \pm 0.2$ | $60.8 \pm 2.0$ | $71.6 \pm 0.7$ | $51.5 \pm 2.7$ | $74.3 \pm 0.7$ |
| Train | $10^3$ | $5.4 \pm 0.2$ | $0.5 \pm 0.2$ | $73.8 \pm 0.3$ | $79.9 \pm 0.3$ | $50.4 \pm 1.9$ | $78.0 \pm 0.5$ |

for evaluation, while the learner continues the continual optimization process. In this scenario, the periodicity $\rho_{\text{eval}}$ is dependent on the evaluator's processing time.

## C    REPRODUCIBILITY DETAILS

**Datasets and transforms.**    For class-incremental learning, Split-MNIST, Split-CIFAR10, and Split-MiniImagenet split their data in 5, 5, and 20 tasks based on their 10, 10, and 100 classes respectively. Each set of data associated to a set of classes then constitutes a task. This split is performed for both the training and test sets, to create a training and evaluation stream of sequential tasks. The order of the class-groupings to create tasks is sequential, e.g. in Split-MNIST we use the digits $\{0, 1\}$ for $T_1$, $\{2, 3\}$ for $T_2$, up to $\{8, 9\}$ for $T_5$. **MNIST** (LeCun & Cortes, 2010) consists of grayscale handwritten digits with ($28 \times 28$) inputs. The dataset contains about 70k images from which 60k training and 10k test images. **CIFAR10** (Krizhevsky et al., 2009) contains 50k training and 10k test images from a range of vehicles and animals with colored ($32 \times 32$) inputs. **Mini-Imagenet** (Vinyals et al., 2016) is a 100-class subset of Imagenet (Russakovsky et al., 2015), with colored inputs resized to ($84 \times 84$). Each class contains 600 images, from which 500 are used for training and 100 for testing. For domain-incremental learning we consider drastic domain changes in **Mini-DomainNet** (Zhou et al., 2021), a scaled-down subset of 126 classes of DomainNet (Peng et al., 2019) with over 90k images, considering domains: clipart, painting, real, sketch. Inputs from all datasets are normalized and Mini-DomainNet resizes the original DomainNet images with their smaller size to 96 using bilinear interpolation, followed by a center crop to fixed ($3 \times 96 \times 96$) inputs. For all datasets, all used transforms are deterministic.

**Gridsearch procedure.**    All experiments for Split-MNIST and Split-CIFAR10 perform a gridsearch over hyperparameters, with each run averaged over 5 initialization seeds. The result with highest ACC is selected as best entry, following (Lopez-Paz & Ranzato, 2017). For computational feasibility in the larger benchmarks, Split-MiniImagenet and Mini-DomainNet, the gridsearch is performed over a single seed and the top entry is averaged over 5 initialization seeds. For all experiments, the learning rate $\eta$ for the gradient-based updates is considered as hyperparameter in the set $\eta \in \{0.1, 0.01, 0.001, 0.0001\}$. A fixed batch size is used for all benchmarks, with 128 for the larger-scale Split-MiniImagenet and Mini-DomainNet, and 256 for the smaller Split-MNIST and Split-CIFAR10. Method-specific and other continual learning hyperparameters are discussed below.

**Continual evaluation setup.** We employ continual evaluation with $\rho_{\text{eval}} = 1$ and subset size 1k per evaluation task, based on our feasibility analysis in Appendix B. For reference with literature, task-transition based metrics ACC and FORG follow standard evaluation with the entire test set.

**Architectures and optimization.** Split-MNIST uses an MLP with 2 hidden layers of 400 units. Split-CIFAR10, Split-MiniImagenet and Mini-DomainNet use a slim version of Resnet18 (Lopez-Paz & Ranzato, 2017; De Lange & Tuytelaars, 2021). SGD optimization is used with 0.9 momentum. Split-MNIST, Split-CIFAR10 and Split-MiniImagenet are configured for 10 epochs per task, which roughly equals to 500, 400, and 200 iterations per task with the defined batch sizes. Mini-DomainNet has 300 training iterations per task to balance compute equally over the tasks.

**Results for identifying the stability gap in Section 4 (intro)** are obtained using Experience Replay (ER) where we consider the loss weighing hyperparameter $\alpha \in \{0.1, 0.3, 0.5, 0.7, 0.9\}$ as in Eq. 8. The above defined batch size indicates the number of new samples in the mini-batch. Additionally, another batch of the same size is sampled from the replay memory, which is concatenated to the new data, following (Chaudhry et al., 2019b; Verwimp et al., 2021). Class-based reservoir sampling is used as in (De Lange & Tuytelaars, 2021) to determine which samples are selected for storage in the buffer $\mathcal{M}$ with a fixed total capacity of $|\mathcal{M}|$ samples. We indicate the selected hyperparameters $(\eta, \alpha, |\mathcal{M}|)$ per dataset here: Split-MNIST $(0.01, 0.3, 2 \cdot 10^3)$, Split-CIFAR10 $(0.1, 0.7, 10^3)$, Split-MiniImagenet $(0.1, 0.5, 10^4)$, Mini-DomainNet $(0.1, 0.3, 10^3)$.

**Results in Section 5.1** use the ER results obtained from Section 4(intro). GEM (Lopez-Paz & Ranzato, 2017) uses the author's standard bias parameter of 0.5, with best learning rates $\eta = 0.01$ and $\eta = 0.001$ for Split-MNIST and Split-CIFAR10 respectively. The task-incremental Split-MNIST setup is based on (van de Ven et al., 2022) with Adam optimizer, batch size 128 and $\eta = 0.001$, but with 200 iterations per task. As it is task-incremental, each task is allocated a separate classifier head and at inference, the corresponding classifier is used for a given sample. Other details remain the same as our class-incremental Split-MNIST setup. The best regularization strength $\lambda$ is chosen based on highest ACC, for SI $\lambda \in \{0.1, 1, \mathbf{10}, 100, 1000\}$, for EWC $\lambda \in \{10^4, 10^5, \mathbf{10^6}, 10^7\}$, for LwF $\lambda \in \{1, 2, \mathbf{5}\}$ and distillation temperature $\{0.5, 1, \mathbf{2}\}$.

**Results in Appendix B** are based on the best ACC results for ER reported in Section 4, see above for details. In the experiment for different subsample sizes, the evaluation and training subsets are obtained from the entire set through a uniform sampling per evaluation.

**Used codebases and equipment.** The experiments were based on the Avalanche framework (Lomonaco et al., 2021) in Pytorch (Paszke et al., 2019). All results were performed on a compute cluster with a range of NVIDIA GPU's.

**Definition of maximal delta increase $\Delta_{t,E_i}^{w,+}$.** We define in the main paper the maximal delta decrease $\Delta_{t,E_i}^{w,-}$, and due to space constraints report the similar definition of maximal delta decrease $\Delta_{t,E_i}^{w,+}$ here in Appendix as:

$$\Delta_{t,E_i}^{w,+} = \max_{m > n} \left( \mathbf{A}(E_i, f_m) - \mathbf{A}(E_i, f_n) \right), \ \forall m, n \in [t - w + 1, \ t] \tag{10}$$

Note that the equation is identical to $\Delta_{t,E_i}^{w,-}$ (Eq. 3), but replacing the constraint $m < n$ with $m > n$.

# D ADDITIONAL RESULTS

## D.1 THE STABILITY GAP IN ONLINE CONTINUAL LEARNING WITH ER

The main results in the paper are based on 10 epochs per task for the class-incremental benchmarks. In *online continual learning*, or streaming continual learning, each sample in the stream is only observed once (Shim et al., 2021; Fini et al., 2020; Chrysakis & Moens, 2020; De Lange & Tuytelaars, 2021; Caccia et al., 2022). We show for Split-MNIST and Split-CIFAR10 in Figure 6 that the stability gap persists for ER in the online continual learning setup.

Additionally, Table 5 shows that our proposed metrics successfully identify the stability gaps in the online continual learning streams. For example in Split-MNIST, the $WF^{10}$ indicates the sheer performance drops of nearly $40\%$ whereas the standard task-based FORG only indicates $5\%$ forgetting. Further, in Split-CIFAR10 an ACC is maintained of $41.8\%$, whereas the min-ACC shows that all tasks drop to zero at least once in the stream.

Figure 6: **Online Continual Learning** only samples each instance from the stream once, resembling a streaming setup. Reports ER accuracy curves for the first four tasks on Split-MNIST (*a-d*) and Split-CIFAR10 (*e-h*), as mean ($\pm$SD) over 5 seeds, with horizontal lines representing average min-ACC. Vertical lines indicate the start of a new task.

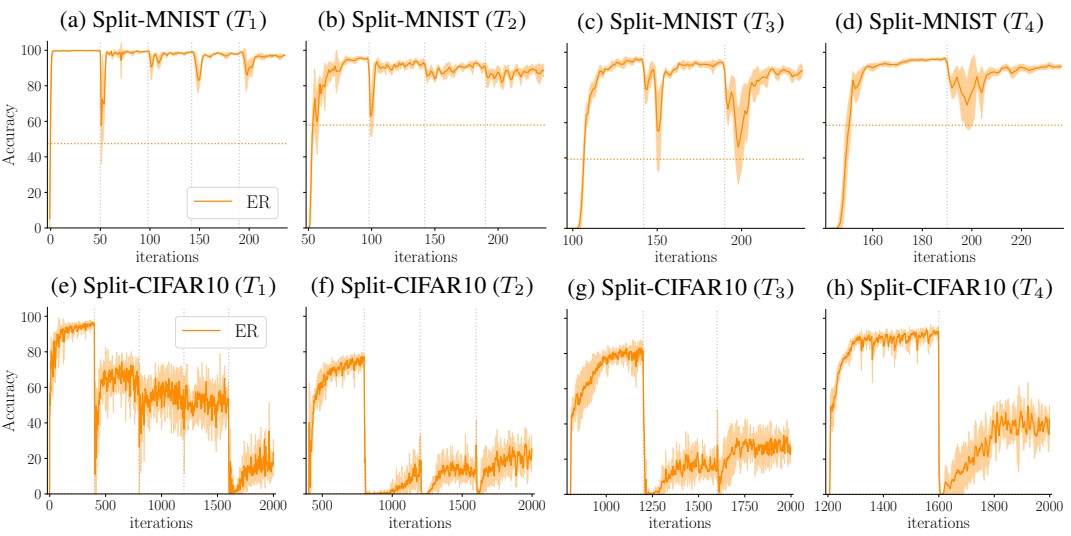

Table 5: **Online Continual Learning results for ER.** Results are reported as mean ($\pm$SD) over 5 seeds.

| | *Trade-off* | | *Stability* | | | | *Plasticity* | |
|---|---|---|---|---|---|---|---|---|
| | **ACC** | **WC-ACC** | **min-ACC** | **FORG** | **WF$^{10}$** | **WF$^{100}$** | **WP$^{10}$** | **WP$^{100}$** |
| Split-MNIST | $91.6_{\pm0.8}$ | $58.9_{\pm7.4}$ | $50.8_{\pm8.7}$ | $5.3_{\pm1.3}$ | $39.9_{\pm7.4}$ | $40.5_{\pm7.5}$ | $86.7_{\pm2.0}$ | $95.8_{\pm1.6}$ |
| Split-CIFAR10 | $41.8_{\pm3.0}$ | $18.2_{\pm0.7}$ | $0.0_{\pm0.0}$ | $55.8_{\pm3.4}$ | $71.5_{\pm4.6}$ | $78.5_{\pm2.8}$ | $55.9_{\pm1.7}$ | $81.4_{\pm2.8}$ |

## D.2 THE STABILITY GAP FOR VARIOUS ER MEMORY SIZES

In the main paper, ER is considered for a single fixed memory size of $M$ samples. Figure 7 shows the class-incremental Split-MNIST results for a range of memory sizes $M \in \{500, 1000, 2000, |S|\}$ where $|S|$ stores all seen samples in the stream. The benchmark setup and ER hyperparameters are identical to Section 4. Three key observations can be made. First, the stability gap persists for all memory sizes, even when storing all samples with $M = |S|$ which approximates joint learning on the observed tasks. Second, with decreasing memory size $M$ the recovery phase deteriorates after the sheer performance drop in the stability gap. This is to be expected as larger memory sizes enable a better approximation of the joint distribution. Third, although storing all samples results in higher overall ACC, Figure 7 indicates a smaller performance drop in the stability gap. This is especially noteable for $T_2$ to $T_4$ right after learning the task when all memory sizes have similar performance. Overfitting on larger memory sizes is more challenging and might take more iterations per task. Therefore, with reference to our conceptual analysis, the gradient norms on task transitions may be higher for larger $M$ and hence decreasing the drop in the stability gap.

Figure 7: ER accuracy curves on Split-MNIST for the first four tasks, for memory sizes $M \in \{500, 1000, 2000, |S|\}$ where $|S|$ stores all seen samples in the stream. Results reported as mean ($\pm$SD) over 5 seeds, with horizontal lines representing average min-ACC. Vertical lines indicate the start of a new task.

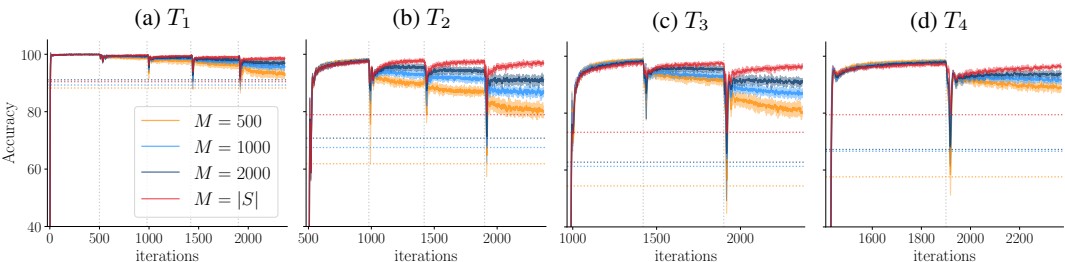

## D.3 WORST-CASE PERFORMANCE ANALYSIS OF FINETUNING

We examine the finetuning method that naively trains on the new task data with stochastic gradient descent (SGD). Finetuning has repeatedly been shown to be prone to catastrophic forgetting (De Lange et al., 2021; van de Ven et al., 2022). We indicate with continual evaluation and our proposed metrics that large forgetting occurs in a window of only a few iterations. We use the same experimental setup with 0.9 momentum as described for the four benchmarks in Section 4, but with plain SGD instead of ER.

Table 6 indicates the catastrophic and prompt forgetting in a window of 10 updates with large values of $\text{WF}^{10}$. Additionally, the worst-case performance min-ACC shows for the class-incremental learning benchmarks that the accuracy declines to zero for previously learned tasks, and on average to $4.9\%$ for the domain-incremental Mini-DomainNet.

Table 6: Finetuning results with continual evaluation for four benchmarks, reported as mean ($\pm$SD) over 5 seeds.

| | Trade-off | | Stability | | | | Plasticity | |
|---|---|---|---|---|---|---|---|---|
| | **ACC** | **WC-ACC** | **min-ACC** | **FORG** | $\textbf{WF}^{10}$ | $\textbf{WF}^{100}$ | $\textbf{WP}^{10}$ | $\textbf{WP}^{100}$ |
| Split-MNIST | $19.5_{\pm 0.1}$ | $19.7_{\pm 0.1}$ | $0.0_{\pm 0.0}$ | $99.7_{\pm 0.1}$ | $86.1_{\pm 3.5}$ | $86.3_{\pm 3.6}$ | $77.8_{\pm 4.4}$ | $96.9_{\pm 3.4}$ |
| Split-CIFAR10 | $17.8_{\pm 0.6}$ | $17.8_{\pm 0.5}$ | $0.0_{\pm 0.0}$ | $85.7_{\pm 0.8}$ | $46.6_{\pm 2.5}$ | $72.4_{\pm 0.6}$ | $62.0_{\pm 1.9}$ | $83.8_{\pm 1.5}$ |
| Split-MiniImagenet | $3.3_{\pm 0.1}$ | $3.3_{\pm 0.1}$ | $0.0_{\pm 0.0}$ | $54.9_{\pm 0.5}$ | $28.4_{\pm 0.8}$ | $52.9_{\pm 0.6}$ | $29.2_{\pm 1.5}$ | $52.8_{\pm 0.8}$ |
| Mini-DomainNet | $15.1_{\pm 1.2}$ | $10.2_{\pm 1.3}$ | $4.9_{\pm 1.3}$ | $13.4_{\pm 1.9}$ | $16.7_{\pm 1.2}$ | $19.6_{\pm 1.1}$ | $10.5_{\pm 0.9}$ | $16.9_{\pm 1.2}$ |

## D.4 THE STABILITY GAP WITH CLASSIFICATION OF SPOKEN WORDS

Figure 8: Per-task accuracy curves (for task 1 to 4) for ER while learning the first five tasks of the speech recognition experiment, performed in a class-incremental manner. Reported is the mean ($\pm$SD) over 5 seeds. Task transitions are indicated by dashed grey vertical lines.

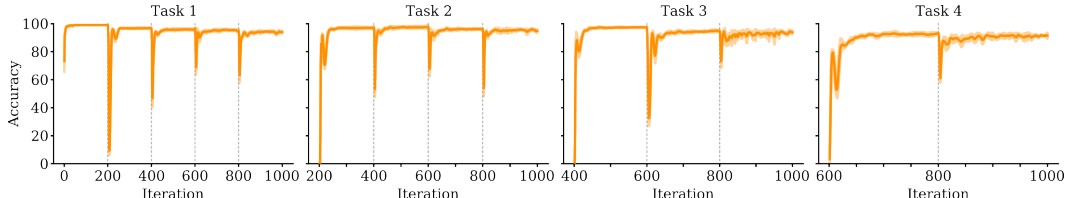

To test whether the stability gap can also be observed in domains other than image classification, we perform a continual learning experiment based on the Synthetic Speech Commands dataset (Buchner, 2017), which contains over 30,000 audio clips of 30 different spoken words. To turn this speech recognition dataset into a continual learning experiment, we closely follow Cossu et al. (2021). The

dataset is split up into different tasks such that each task contains all the training samples of two words, and these tasks are presented to the algorithm one after the other. We use the same sequence of tasks as Cossu et al. (2021). The experiment is performed according to the class-incremental learning scenario (i.e., task identities are not provided at test time), so the algorithm must learn to distinguish between all words encountered so far.

Following Cossu et al. (2021), we used an MLP with a single hidden layer containing 1024 units with ReLU activations, followed by a softmax output layer. Training was done using the Adam optimizer with learning rate 0.0001. Each task was trained for 200 iterations with mini-batch size 256. We used the same preprocessing steps as Cossu et al. (2021): for each audio clip 40 Mel coefficients were extracted using a 25 ms sliding window with 10 ms stride, resulting in fixed-length sequences with 101 time steps. These sequences where aggregated over time, so that each sample fed to the network consisted of $40 \times 101 = 4040$ input features.

On this class-incremental learning experiment we tested the method ER, using a memory budget of 100 examples per class. Figure 8 displays the per-task accuracy curves for the first five tasks, displaying clear stability gaps after every task transition, thus confirming that the stability gap phenomenon is not specific to the domain of image classification.

### D.5 CORRELATION CATASTROPHIC FORGETTING AND STABILITY GAP METRICS

Figure 9: Correlation analysis between task-based forgetting (catastrophic forgetting measure) and windowed-forgetting (stability gap measure). After each newly learned task, the FORG and $WF^{10}$ are included for all evaluation tasks. Pearson correlation coefficient $\rho$ shows strong correlation for easier benchmark Split-MNIST, but is closer to zero for the more challenging benchmarks. Correlation is determined on the averages over 5 seeds (bold entries), light entries illustrate per-seed results.

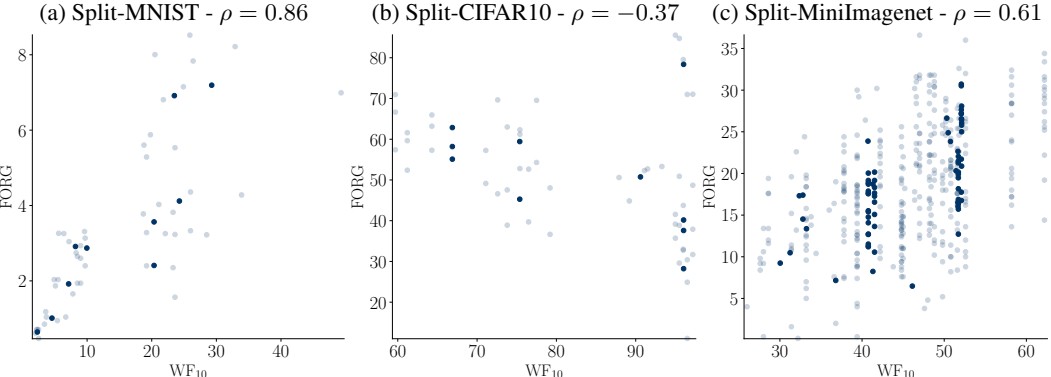

Two of our proposed metrics directly measure properties of the stability gap. $WF^{10}$ measures how steep the initial relative drop is after learning a task and min-ACC measures the absolute minimum performance. With a correlation analysis we hope to gain additional insights w.r.t. the standard measure for catastrophic forgetting (FORG).

On each task transition in the learning stream, we compare the FORG with the $WF^{10}$ for each of the evaluation tasks separately. This approach can compare per evaluation task how steep the stability gap is, and what the effect is for catastrophic forgetting. The experiment considers evaluation data for ER from the analysis in Figure 2 in the main paper. We also considered the min-ACC as measure for the stability gap, but as it often drops to zero accuracy after the first task, it doesn't provide further insights.

Figure 9 indicates results for our correlation study on Split-MNIST, Split-CIFAR10, and Split-MiniImagenet. The easier Split-MNIST shows a high Pearson correlation coefficient $\rho = 0.86$. However, the linear relation is less prominent for the more challenging Split-CIFAR10 and Split-MiniImagenet with $\rho = -0.37$ and $\rho = 0.61$. For the latter two, we specifically observe formation of columns. As the $WF^{10}$ measures a maximum drop for the evaluation task due to the stability gap, the accuracy can still recover with ER when converging for the new task. However, over time when learning more tasks in the stream, the accuracy at recovery of the evaluation task declines. This

results in entries with constant WF[10], whereas FORG declines. This relation between the two metrics can also be qualitatively observed in the accuracy curves of Figure 2 in the main paper.

## D.6 ADDITIONAL GEM RESULTS FOR SPLIT-MNIST AND SPLIT-CIFAR10

The main paper shows results for Split-MNIST accuracy curves and zooms of the stability gradient norms on task transitions. In Figure 10 we provide the full $\nabla \mathcal{L}_{\text{stability}}$ norm results during the learning process. For the analysis of the stability gap, the zoom on the task transitions is most informative, whereas the full results give an overview of gradient magnitude towards convergence of the tasks. GEM and ER are reported for the best ACC, both with learning rate 0.01 (see details in Appendix above). Compared to GEM, ER exhibits larger stability gradient norms over nearly the entire learning trajectory. This indicates that towards convergence of the task, the mini-batch gradients mostly satisfy the GEM constraints with gradient angles $\leq 90°$, hence resulting in projection vectors with small magnitude. Although using the gradients of the replay samples directly for ER converges to low gradient norms, there is a notable difference for the near-zero values for GEM where these gradients are used for angle-based constraints.

Additionally to the results on Split-MNIST, Figure 11 provides the accuracy curves for the first four tasks on Split-CIFAR10. These results are in correspondence with literature (De Lange & Tuytelaars, 2021; Aljundi et al., 2019b;a), where GEM is not effective for the more difficult class-incremental Split-CIFAR10 benchmark. Notable is the sharp drop to near-zero accuracy on task transitions (stability gap), which subsequently peaks and then declines rapidly.

Figure 10: GEM and ER per-iteration L2-norms of current $\nabla \mathcal{L}_{\text{stability}}$ on Split-MNIST. Results reported as mean ($\pm$SD) over 5 seeds. Vertical lines indicate the start of a new task.

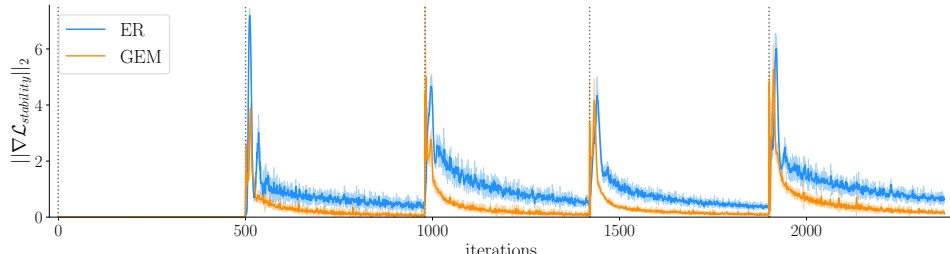

Figure 11: GEM and ER accuracy curves for the first four tasks of Split-CIFAR10, reported as mean ($\pm$SD) over 5 seeds. The min-ACC averaged over seeds is zero for both methods.

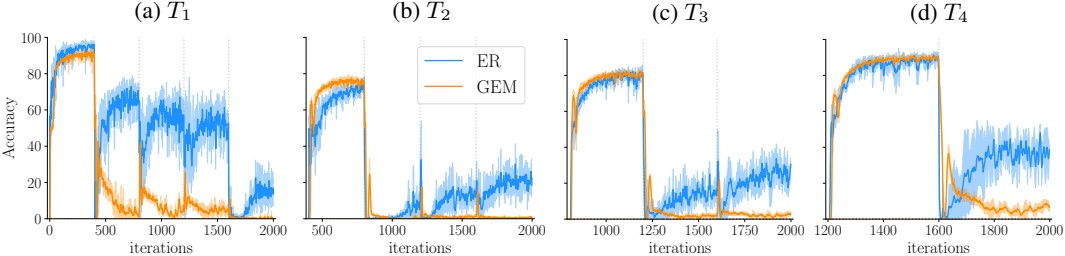

## D.7 ADDITIONAL LWF RESULTS FOR MINI-DOMAINNET

Besides the empirical analysis on Split-MNIST and Rotated-MNIST, we demonstrate the stability gap for Learning without Forgetting (LwF) (Li & Hoiem, 2017) in the domain-incremental Mini-DomainNet (Figure 12). The stability gaps are especially notable for $T_1$ and $T_2$ in Figure 12(a,b). Figure 12(e) reports the stability gradient norms, with a focus on the task-transitions in Figure 12(f,g). The results confirm the zero gradient norms on task transitions, followed by recovery with increasing gradient norms.

Figure 12: LwF and ER accuracy curves (*a-d*) on Mini-DomainNet for the four domains, and the per-iteration L2-norms of current $\nabla\mathcal{L}_{\text{stability}}$ (*e-g*). Results are reported as mean ($\pm$SD) over 5 seeds, with horizontal lines representing average min-ACC. Vertical lines indicate the start of a new task. Note that the x-axis scale varies over (*a-d*), (*e*) overviews the full learning trajectory, and (*f,g*) are zooms in on the first few iterations for the last two tasks.

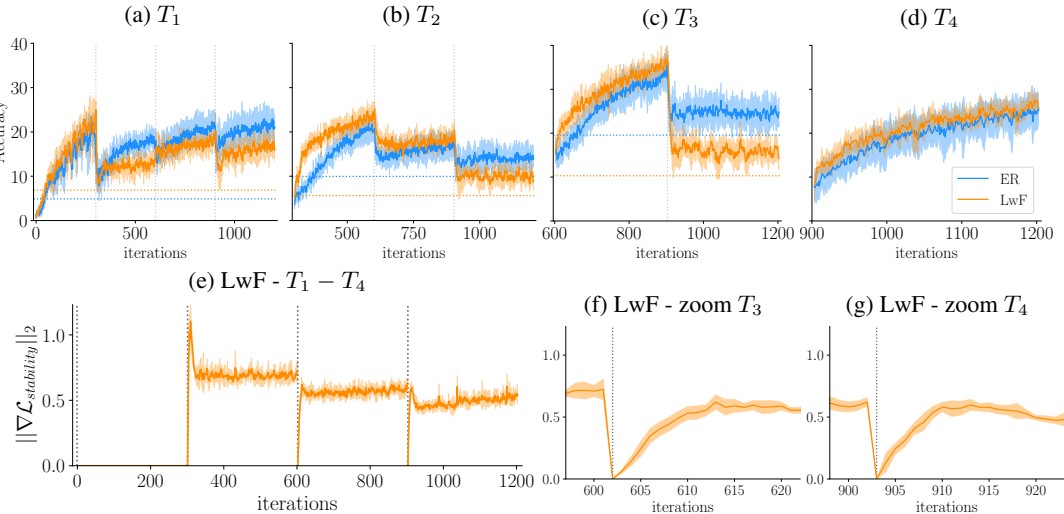

