# OpenReview forum: "Continual evaluation for lifelong learning: Identifying the stability gap"
_ICLR.cc/2023/Conference — ICLR 2023 notable top 25%_

### Official Review · Reviewer_Jyqa · 2022-10-20

**Confidence:** 5
**Correctness:** 3
**Technical Novelty And Significance:** 3
**Empirical Novelty And Significance:** 2
**Recommendation:** 5

**Clarity, Quality, Novelty And Reproducibility:**

Good quality, but the finding is a little bit trivial and limit to  replay-based methods.
Nice clarity.
Nice originality.

**Strength And Weaknesses:**

Strengths:

(a) This paper is well-written and easy to read.

(b) The phenomenon found in this work is easy to follow.

Weaknesses:

(a) Some notations are confusing. What  is $|T_{i}|$? In Section 3.2, what is the maximal accuracy increase (lack of detailed formulations)?

(b) My main concern is about the phenomenon---stability gap. This work claims: ``we show that common state-of-the-art methods still suffer from substantial forgetting upon starting to learn new tasks, except that this forgetting is temporary and followed by a phase of performance recovery.'' I am afraid that this phenomenon can not cover most kinds of baselines in CL and only exists in replay-based methods. The phenomenon is trivial in replay-based methods and easy to be explained. The model reaches at a local minimum in the previous task and tries to approach to next task's local minimum quickly in the early several iterations, which leads to the substantial forgetting upon starting to learn new tasks. As replay-based methods replay previous experience, the model learns previous tasks' skills in the following iterations. These lead to the phase of performance recovery. The baselines  discussed in this paper is mainly replay-based (ER and GEM), and the phenomenon in LwF is not well explained.  I think this phenomenon is more likely to not exist in other kinds of baselines---regularization-based (EWC, SI and MAS) and parameter isolation based (PackNet and HAT). For example, the work [1] plots a similar accuracy curve of EWC in Fig.3(a) where there is no stability gap.

[1] Kirkpatrick J, Pascanu R, Rabinowitz N, et al. Overcoming catastrophic forgetting in neural networks[J]. Proceedings of the national academy of sciences, 2017, 114(13): 3521-3526.

**Summary Of The Paper:**

This paper try to show that common state-of-the-art methods in continual learning suffer from substantial forgetting upon starting to
learn new tasks, except that this forgetting is temporary and followed by a phase of performance recovery. The main contributions can be summarized as :

(1) This work defines a framework for continual evaluation that evaluates the learner after each update.

(2) This work conducts an empirical study with the continual evaluation framework, which leads to identifying the stability gap for Experience Replay.

(3) This work proposes a conceptual analysis as a hypothesis for causing the stability gap, by disentangling the gradients based on plasticity and stability.

**Summary Of The Review:**

marginally below the acceptance threshold

---

> ### Public Comment · ~Lucas_Caccia1 · 2022-11-11
> **On why we observe the stability gap**
>
> I want to first thank the authors for shining a light on an important issue, and specifically on how standard evaluation protocols are blind to this phenomenon. I think the field can greatly benefit from better evaluation metrics.
>
> I would like to discuss the root cause of this stability gap. I think this problem does not stem from the method being used, but rather from the learning problem itself. In our ICLR paper [1] last year, we investigated what exactly happens at a task switch in the class-incremental setting. In brief, our findings were that:
>
> 1. When new classes are introduced in a new task, the model initially projects them close to the prototypes (rows in the last linear layer) of previous classes
> 2. Rather than shifting new points away from previous classes, a standard cross-entropy loss instead pushes the old prototypes away from the new points
> 3. After a few gradient steps, the prototypes of previous classes are miscalibrated, with the model predicting that older points belong to new classes.
>
> We also found that directly addressing this issue solves the stability gap (please see Fig. 1).
>
> To my surprise, the authors also found that this issue can occur in domain incremental settings, such as rotated mnist where no new class prototypes are added. This is a very interesting finding, and it would be good to check whether or not a similar prototype miscalibration phenonemon is occuring there too : for example, can you find classes $a,b$  where the hidden representations of class $a$ at task $i$ strongly overlaps with class $b$ at task $i+1$ ?
>
> Finally, it would be very interesting to evaluate the stability gap in multi-head settings, with training performed such that classes from different tasks do not interact with each other. I think this would give us good insight on the subtleties of the stability gap.
>
>
> Thank you.
>
> [1] https://openreview.net/pdf?id=N8MaByOzUfb

---

> > ### Comment · Reviewer_Jyqa · 2022-11-11
> > **Thank you for response.**
> >
> > Thank you for your response. It is glad to discuss the interesting phenomenon---stability gap---in this paper.
> >
> > (a) I argue that this phenomenon is more likely to only exist in replay-based methods. I think it would be better to demonstrate whether it exists in other kinds of baselines (regularization-based or parameter isolation based) by theoretical or intuitive analysis.
> >
> > (b) In the response, authors try to discuss the phenomenon in terms of the learning problem itself, and discuss it in three settings (or scenarios). In fact, authors spend a lot of pages analyzing the phenomenon in terms of methods in Section 5 (CONCEPTUAL ANALYSIS OF THE STABILITY GAP) and Section 5.1 (named as FINDING A STABILITY GAP IN OTHER CONTINUAL LEARNING METHODS). If authors want to analyze this phenomenon in terms of settings, great efforts about settings need to be spent in the main file instead of plenty discussion about methods.
> >
> > (c) I'm not trying to prove that the phenomenon is worthless. Instead, I think it is interesting. I just think the phenomenon is limit to the replay-based methods and easy to be explained, which makes it seem to be not very significant.
> >
> > Best wishes.

---

> > > ### Public Comment · ~Lucas_Caccia1 · 2022-11-11
> > > **Clarification**
> > >
> > > Good observation, yes I agree that this would not be an issue in parameter isolation methods (because such methods can recover exactly the model after the first task, so there would be no change in performance on the first task).
> > > However, I think the stability gap issue would be as severe (if not worse) for a finetuning baseline, since it would still face the issue I tried to highlight above. I do think that regularization based methods could be more robust to this, simply because they restrict the model from moving too much from a previous solution.
> > >
> > > FYI, **I am not an author on this paper**, so I don't see why sharing relevant work would be problematic :)

---

> > > > ### Comment · Reviewer_Jyqa · 2022-11-12
> > > > **Thank you for your response**
> > > >
> > > > Thank you for your response. I am sorry that I misidentified you as the author, and I have revised my comments. It is nice to have a free discussion at ICLR. Yes, I do think that there is no clear phenomenon in regularization-based methods. The substantial forgetting is an important part in this phenomenon, which is shown in this paper. In regularization-based methods, restricting the model from moving too much from a previous solution may not directly result in the substantial forgetting.
> > > >
> > > > Best wishes.

---

> > ### Author Response · Authors · 2022-11-18
> > **Author's response to open comment**
> >
> > We thank you for the comment and interesting discussion. We agree that further studying the root causes of the stability gap is an important and interesting direction for future work. The work that you refer to could be considered a first step in that direction, but as you already indicate this explanation does not cover the domain- or task-incremental settings, while we show that the stability gap can be observed in these settings as well (see also several newly added experiments in the rebuttal).
> >
> > We would also like to point out that the stability gap we describe in this paper is not only the sudden drop in performance on past tasks after a task transition, but also the later spontaneous recovery of past task performance. We consistently observe this forget-and-recover behavior in our experiments for replay, constraint-based replay, distillation-based, and regularization-based methods.

---

> ### Author Response · Authors · 2022-11-18
> **Answer (1/2): New results for LwF, EWC and SI in task and domain-incremental learning**
>
> We thank the reviewer for the valuable feedback and respond to the raised concerns in the following. Due to space constraints, we respond in two comments with references in the final one.
>
> **“The phenomenon is trivial in replay-based methods and easy to be explained. [...] The baselines discussed in this paper is mainly replay-based (ER and GEM), and the phenomenon in LwF is not well explained.”**
>
> We agree that with the hindsight of observing the stability gap, sensible explanations can be found for why the stability gap affects ER. We believe this makes the finding even more interesting as ER is a commonly used SOTA method, and we are the first to analyze this consistent behavior of a sheer drop in performance and subsequent recovery.
> Additionally, even though GEM is a replay method, it is a \emph{constraint-based} replay method. This indicates that samples are not actually relearned, but their gradients are used to constrain the current gradient. The samples are hence not simply relearned, and interestingly even for this method the stability gap is still observed.
>
> As per the reviewer’s suggestions, we rephrased our conceptual analysis for LwF for clarity. On top of the LwF results with a stability gap in Mini-Domainnet (now in Appendix D.6), we report additional LwF results in Section 5.1 for task-incremental Split-MNIST and the domain-incremental Rotated-MNIST. The results and discussion can be found in the revised version of the paper, with changes indicated in blue.
>
>
> **“I am afraid that this phenomenon can not cover most kinds of baselines in CL and only exists in replay-based methods. [...]  I think this phenomenon is more likely to not exist in other kinds of baselines---regularization-based (EWC, SI and MAS) and parameter isolation based (PackNet and HAT).”**
>
> *Regularization based methods*
>
> We empirically tested the reviewer’s hypothesis and found that the stability gap can also be observed with regularization-based methods (EWC and SI) and with a distillation-based method (LwF). In Section 5.1 we conduct experiments for EWC, SI and LwF both on the task-incremental and domain-incremental setting, where these methods can still be competitive w.r.t. ER [4]. More specifically, we observe the stability gap in task-incremental Split-MNIST, where each task is allocated a separate classifier head as in [5], and domain-incremental Rotated-MNIST with 80 degrees of rotation on the images between subsequent tasks.
> We have also added a conceptual analysis for EWC and SI, and we have updated our conceptual analysis for LwF in Section 5.1. All regularization methods share a common idea: stay close to the previous task solution. However, on the first updates when learning a new task, the solution is still very similar, and hence the regularization objective is negligible. For the first update on the new task, the parameters (for EWC,SI) and the model’s input-output mapping (for LwF) are still *identical* to the previous task model, hence making the regularization term exactly zero. The full discussion can be found in Section 5.1 in the updated version of the paper.
>
> *Parameter-isolation based methods*
>
> We agree with the reviewer on the comment about parameter isolation based methods (PackNet and HAT). In fact, these methods are not considered in our empirical study for the stability gap, as typically fixed parts of the model are allocated to specific tasks [1-3]. The task identifier is typically assumed available for inference, allowing to activate only the subnetwork for that task. This allows to prevent any changes (e.g. by masking) and hence forgetting for observed tasks. Therefore, for methods where no changes in performance of learned tasks is allowed, a stability gap cannot be present either. For this rebuttal, we added a discussion on parameter isolation methods in Appendix A (see revised paper version with changes in blue).
>
> **“For example, the work [1] plots a similar accuracy curve of EWC in Fig.3(a) where there is no stability gap.”**
>
> Even though at first glance it seems that in these plots there is no stability gap, if you look closely the performance of EWC does appear to have small stability gaps (i.e., small dips in the top red curve upon starting to train on Task B and on Task C, and a small dip in the middle red curve upon starting to train on Task C). Although we certainly agree that from this graph the stability gap is not very convincing. This can however have several reasons. For example, it is unclear what the evaluation periodicity is in these graphs.
> We therefore ran a similar experiment with EWC on permuted MNIST ourselves, and in this experiment we found clear stability gaps. This further strengthens our observation that EWC is affected by the stability gap as well.
> A graph with the per-task accuracy curves, along with a description of the experimental details, can be found in this anonymized repository: https://anonymous.4open.science/r/EWConPermutedMNIST-393B

---

> > ### Author Response · Authors · 2022-11-18
> > **Answer (2/2): Notation and scope improvements**
> >
> > **“Some notations are confusing. What is $|T_i|$? In Section 3.2, what is the maximal accuracy increase (lack of detailed formulations)?”**
> >
> > Our apologies for the confusion in notation. We elaborate in the following on our changes to improve clarity in the revised paper.
> >
> > - In the previous version of our paper, $|T_k|$ was meant to indicate the overall iteration number of the last training iteration of task $T_k$. To improve clarity of the notation, we significantly rephrased Section 2: we elaborated on the meaning of the overall iteration number ‘$t$’ and altered the notation throughout the paper from $|T_k|$ to  $t_{|T_k|}$ to more clearly indicate that this refers to the overall iteration number (rather than for example the number of iterations within task $T_k$ itself).
> >
> >
> > - We have clarified the definition of the maximal accuracy increase in section 3.2, by pointing out its similarity to the maximal accuracy decrease in Eq.3 and by providing the full equation in the Appendix (Eq.10).
> >
> > All changes in notation are indicated in blue in the updated version of the paper in OpenReview.
> >
> > **“This work claims: ``we show that common state-of-the-art methods still suffer from substantial forgetting upon starting to learn new tasks, except that this forgetting is temporary and followed by a phase of performance recovery.'' I am afraid that this phenomenon can not cover most kinds of baselines in CL and only exists in replay-based methods.”**
> >
> > We refer to the above discussion where we report results confirming the stability gap for ER, GEM, LwF, EWC, and SI, and where we discuss how parameter isolation methods by design cannot have a stability gap.
> > Further, to increase clarity and to better outline the scope, in the abstract we rephrased our use of *‘common state-of-the-art methods’* to *‘a set of common state-of-the-art methods’*.
> >
> >
> > **References**
> >
> > [1] Mallya, Arun, and Svetlana Lazebnik. "Packnet: Adding multiple tasks to a single network by iterative pruning." Proceedings of the IEEE conference on Computer Vision and Pattern Recognition. 2018.
> >
> > [2] Serra, Joan, et al. "Overcoming catastrophic forgetting with hard attention to the task." International Conference on Machine Learning. PMLR, 2018.
> >
> > [3] Rusu, Andrei A., et al. "Progressive neural networks." arXiv preprint arXiv:1606.04671 (2016).
> >
> > [4] Van de Ven, Gido M., and Andreas S. Tolias. "Three scenarios for continual learning." arXiv preprint arXiv:1904.07734 (2019).
> >
> > [5] De Lange, Matthias, et al. "A continual learning survey: Defying forgetting in classification tasks." IEEE transactions on pattern analysis and machine intelligence 44.7 (2021): 3366-3385.

---

### Official Review · Reviewer_FwH6 · 2022-10-24

**Confidence:** 4
**Correctness:** 4
**Technical Novelty And Significance:** 3
**Empirical Novelty And Significance:** 3
**Recommendation:** 8

**Clarity, Quality, Novelty And Reproducibility:**

In terms of quality and novelty, I find the work decent. In addition, the paper is well-organized and easy to follow. Regarding the reproducibility, the authors provide comprehensive details of their setup in the paper.

**Strength And Weaknesses:**

### Strengths
- The stability gap is an interesting problem in continual learning.
- The provided metrics are helpful for some sensitive scenarios.
- The experiments are well-desigend.


### Weakness
- The authors do not propose a mechanism to alleviate the stability gap (and potentially catastrophic forgetting). However, I believe as a first step, the analysis of Sec. 5 is enough to motivate others to work on this problem.




**Summary Of The Paper:**

The work is motivated by a problem that current continual learning metrics are mainly coarse-grained (e.g., task-based), which can lead to information loss. For instance, monitoring task-based metrics cannot show the stability gap (substantial but temporary forgetting upon learning a new task). The stability gap is defined as the sudden drop in performance right after facing a new task.


Further, the authors propose various metrics for continual learning that focus on worst-case performance, which can be helpful in practical scenarios. Finally, the paper briefly studies the implications of the stability gap from the task similarity perspective. Also, it shows that this phenomenon exists in the presence of other CL algorithms.

**Summary Of The Review:**

Overall, While there is room for improvement in a more in-depth analysis of the stability gap, I find this work a good first step that can contribute significantly to the field. Hence, I recommend acceptance.

---

> ### Author Response · Authors · 2022-11-18
> **Additional results in this rebuttal**
>
> *“Overall, While there is room for improvement in a more in-depth analysis of the stability gap, I find this work a good first step that can contribute significantly to the field.”*
>
> We thank the reviewer for the kind words and insightful review, and share the belief that our findings could motivate further investigation into the stability gap. Additionally, for this rebuttal, we added extensive additional empirical evidence, which we hope has further enhanced the depth of our analysis. In summary:
> - Besides existing results for GEM, ER, and LwF, we additionally predict in our conceptual analysis and demonstrate in empirical experiments that the stability gap also affects regularization-based methods EWC and SI.
> - We go beyond the vision domain and show that the stability gap can also be observed with speech classification (Appendix D.8).
> - We show for ER that our findings persist over multiple memory sizes, and find that even when all samples are stored (i.e., approximating sequential ‘joint’ training) the stability gap is still observed (Appendix D.2).
> - Besides class and domain incremental learning, we extend the scope of our experiments to task-incremental learning (Section 5.1), and online Continual Learning (Appendix D.1).

---

### Official Review · Reviewer_xcu2 · 2022-10-25

**Confidence:** 2
**Correctness:** 3
**Technical Novelty And Significance:** 3
**Empirical Novelty And Significance:** 4
**Recommendation:** 8

**Clarity, Quality, Novelty And Reproducibility:**

Overall, the work and the proposed concept of stability gap is novel and can be beneficial in measuring the forgetting and gaining new information in continual learning.

**Strength And Weaknesses:**

The idea of measuring the stability-gap which shows how much information is retained and how much new information is learnt after each iteration is a very interesting concept in continual learning since  by learning new tasks the model tries to recover its knowledge after losing it in first timestamps for that task.
The paper strengths are:
1) The good explanation of the problem and the solution.

2) Proposing a concept that tries to measure the tradeoff between forgetting and getting new knowledge after each iteration.

3) The analysis of the proposed stability gap and its dependency with tasks' similarity make the idea more promising.

Some weaknesses that I can mention for this work are:

1) The idea and analysis have been implemented only for image-based datasets while the continual learning happens for other domains such as texts. The shortage of experiments and analysis of the metric on such datasets should be addressed to show the generalizability of the stability gap for different domains.

2) To measure the relation between stability gap and tasks differences, a synthetic dataset is created by rotating images, however studying this relation on more real-cases tasks can be more beneficial and important to show the relevance.

This is a minor question and some explanations can make it more clear: according to Table 1 stability gap changes by increasing the evaluation periodicity, therefore what is the best periodicity that should be used in such evaluations, is it domain dependent?



**Summary Of The Paper:**

This paper proposes stability gap as a concept to measure the transient forgetting that happens for continual learning. It measures the trade-off between the knowledge conservation by learning new tasks (retaining knowledge) and getting new knowledge for the new task. The relation of stability gap with similarity between tasks has bee explored and its analysis also has been conducted.

**Summary Of The Review:**

This paper proposes a concept that tries to measure the tradeoff between forgetting and getting new knowledge in continual learning which is very important in such learnings. The analysis of this concept also helps to better understand it. Overall motivation and idea and experiments are convincing, the only concern is its analysis that is only limited to image-based datasets and some of the experiments are conducted for synthetic datasets that are not close to real-case scenarios.

---

> ### Author Response · Authors · 2022-11-18
> **New results beyond vision and clarifications**
>
> We thank the reviewer for the valuable feedback. In the following, we address the two mentioned weaknesses and the question of the reviewer.
>
> **W1: “only for image-based datasets while the continual learning happens for other domains”**
>
> Our study focused on the vision domain as this is a typical standard in the field, for reference we refer to these surveys [1,2,3]. However, we agree with the reviewer that testing whether the stability gap can also be observed in domains other than image classification is important, and would strengthen the paper. We have therefore added an additional experiment (see Appendix D8) using a speech recognition dataset with audio clips of spoken words. With this dataset we construct a continual learning experiment in which an algorithm must incrementally learn to distinguish between up to 30 different spoken words. Promisingly, also on this benchmark, we find substantial stability gaps after every task transition, thus confirming that the stability gap phenomenon is not specific to the domain of image classification.
>
>
> **W2: “To measure the relation between stability gap and tasks differences, a synthetic dataset is created by rotating images, however studying this relation on more real-cases tasks can be more beneficial and important to show the relevance.”**
>
> The reviewer refers with rotating task-images to the Rotated-MNIST experiment.  This experiment was deliberately a controlled experiment designed to regulate the task similarity. We agree that this is not a real-world scenario, but as task similarity is hard to measure for real-world datasets, Rotated-MNIST enables an interpretable and controllable alternative.
>
> Moreover, we have reported extensive empirical evidence on 7 benchmarks for class-incremental learning (Split-MNIST, Split-CIFAR10, Split-MiniImagenet, the new speech recognition experiment), domain-incremental learning (Rotated-MNIST and Mini-DomainNet), and now also task-incremental learning (multi-head Split-MNIST). We agree with the reviewer that challenging and real-world benchmarks are important to show the relevance of our findings, this is why we incorporated Split-MiniImagenet for the class-incremental benchmarks, i.e. a 100-class subset of Imagenet (Russakovsky et al., 2015), and Mini-DomainNet (Zhou et al., 2021), a scaled-down subset of 126 classes of DomainNet (Peng et al., 2019) with over 90k images, considering domains: clipart, painting, real, sketch.
> Additionally in this rebuttal, we show with a new speech recognition experiment that our findings hold beyond vision.
>
> **Q1: “according to Table 1 stability gap changes by increasing the evaluation periodicity, therefore what is the best periodicity that should be used in such evaluations, is it domain dependent?”**
>
> We thank the reviewer for the question and can confirm that the stability gap changes as more coarse evaluation periodicity is used. Figure 2 in the main paper visualizes this phenomenon, and this is exactly why the stability gap remained under the radar using the commonly used task-based evaluation periodicity. Therefore, the most precise evaluation for identifying the stability gap is always the per-step evaluation. Depending on the application, coarser evaluation may be used for computational efficiency.
>
> We explore efficient alternatives for tractable continual evaluation in Appendix B, exploring evaluation periodicity, evaluation set subsampling, and discuss practical considerations for real-world application. Appendix B indicates that subsampling the evaluation sets is a better strategy to reduce compute, while retaining a representative estimate of the minimum and worst-case ACC.
>
>
> **References**
>
> [1] De Lange, Matthias, et al. "A continual learning survey: Defying forgetting in classification tasks." IEEE transactions on pattern analysis and machine intelligence 44.7 (2021): 3366-3385.
>
> [2] Parisi, German I., et al. "Continual lifelong learning with neural networks: A review." Neural Networks 113 (2019): 54-71.
>
> [3] Masana, Marc, et al. "Class-incremental learning: survey and performance evaluation on image classification." arXiv preprint arXiv:2010.15277 (2020).

---

> > ### Comment · Reviewer_xcu2 · 2022-11-29
> > **Addressed Suggestion**
> >
> > Thanks authors for providing the answers and also applying the method to the new dataset. I changed my score due to the new conducted experiments on speech recognition task that shows the generalizability of the method on other domains.

---

### Official Review · Reviewer_q7YA · 2022-10-27

**Confidence:** 5
**Correctness:** 4
**Technical Novelty And Significance:** 3
**Empirical Novelty And Significance:** 3
**Recommendation:** 8

**Clarity, Quality, Novelty And Reproducibility:**

The paper is well-written and easy to understand. A sufficient ablation analysis is provided to support the claims. It provides a novel framework for evaluating continual learning approaches and the significance of the proposed evaluation framework in a continual learning setup. The authors have assured us they will share the code publicly after acceptance.

**Strength And Weaknesses:**

Strengths:
1- The paper identifies the issues that occur in the existing task
incremental continual learning approaches- and tries to address them by providing empirical experiments and detailed analysis.

2- It proposes a framework for continual evaluation that evaluates the learner after each update and provides an ablation study to prove why evaluation frequency indicates continual evaluation is necessary to surface the stability gap.

3- It provides an empirical study with the continual evaluation framework to identify the stability gap for Experience Replay.

4- To quantify the stability gap in contrast to existing metrics in the existing approaches, it proposes novel metrics such as the minimum and worst-case accuracy (min-ACC and WC-ACC). It shows that the stability gap is significantly influenced by the degree of similarity of consecutive tasks in the data stream.

5-  It also provided a conceptual analysis for causing the stability gap and conducted the experiments for several existing approaches such as Experience Replay (Chaudhry et al., 2019b), GEM (Lopez-Paz & Ranzato, 2017), and LwF (Li & Hoiem, 2017) for supporting the proposed hypothesis.

Weaknesses:
1- In the experimental setup for MNIST, as you mentioned (on page 6), "each task constitutes the entire MNIST dataset." Is it mean that each task consists of all class examples (0,1...9)? If yes, then how is it valid for continual learning? Because there should be new class examples in the next task. How a buffer capacity affects ER performance?
2- How your new evaluation setup work for online continual learning approaches such as [a],[b] and[c]? Is the newly proposed metrics (worst case and Min) helpful in identifying this stability gap?
3- The extreme case for continual learning is where the data examples are available only once, unlike the replay-based approaches, where the old data may be available for future training [d]. It would be interesting to apply the proposed framework to this challenging continual setup [d].


[a]- Online Class-Incremental Continual Learning with Adversarial Shapley Value, AAAI-2021.
[b]- Online Continual Learning under Extreme Memory Constraints, ECCV 2020.

[c]- Online continual learning from imbalanced data, ICML 2020.
[d]- Efficient feature transformations for discriminative and generative continual learning, CVPR 2021.










**Summary Of The Paper:**

This paper pointed out the shortcomings in the existing approaches for continual learning, mainly catastrophic forgetting of the previous task during new task adoption. It proposes per-iteration continual evaluation with new metrics that enable measuring worst-case performance. This work empirically studied five existing continual works and shows that they suffer from a significant performance loss on previous tasks. The stability gap increases significantly as subsequent tasks are more dissimilar. It also explains the stability-plasticity trade-off by proposing a new metric evaluation for continual learning. The conceptual analysis is provided for causing the stability gap by disentangling the gradients based on plasticity and stability.











**Summary Of The Review:**

The paper has pointed out the interesting shortcomings in the existing continual learning approaches and proposed a new evaluation framework to address them. I encourage the authors to include my concerns raised in the weaknesses subsection. For more detail, please see the strength and weaknesses section.

---

> ### Author Response · Authors · 2022-11-18
> **Answer(1/2): new results for both online CL and ER memory sizes; and clarifications**
>
> We thank the reviewer for the thoughtful review. Below we address the weaknesses mentioned by the reviewer and due to space constraints add the corresponding references in a second comment.
>
> **Q1a: Does each task constitute the entire MNIST dataset?**
>
> On page 6 (Section 4.1) the Rotated-MNIST experiment indeed comprises the entire MNIST dataset per task, and each task hence comprises all 10 classes.
> However, key to this benchmark introduced by [1] is that for each task a different (but fixed!) rotational transform is applied. This is how we control the task similarity.
> A benchmark where only the input domain changes, is called a domain incremental learning setup [2]. We have rephrased the experiment setup in Section 4.1 to further clarify this benchmark. Additionally, we adapted all *class-incremental* benchmark names to set them more clearly apart from the *domain incremental* ones, changing MNIST, CIFAR10 and Mini-Imagenet to Split-MNIST, Split-CIFAR10, and Split-MiniImagenet. Changes are indicated in blue in the updated paper version.
>
>
> **Q1b: How does buffer capacity affect performance of ER?**
>
> We thank the reviewer for the suggestion of this interesting experiment. In a newly added experiment, we now evaluate the performance of ER in Split-MNIST for buffer capacities $M=\\{0.5k, 1k, 2k, |S|\\}$ in Appendix D.2, where $|S|$ denotes ER with unlimited buffer capacity, hence resembling joint training. Interestingly, the stability gap persists over all considered M, even with unlimited capacity. Consistent over the results is that for increasing M, better recovery occurs after the drop of the stability gap. This is to be expected as the ER buffer better estimates the joint data distribution. For the full analysis we refer to Appendix D.2.
>
> **Q2: How does the new evaluation setup work for online continual learning [a-c]? Are the newly proposed metrics [...] helpful in identifying this stability gap?**
>
> Besides continual learning experiments with multiple epochs per task, we now also include results for online continual learning, where only one epoch per task is allowed as in a streaming setting. Promisingly, both on Split-MNIST and Split-CIFAR10 we find that the stability gap persists for ER. The new results can be found in Appendix D.1 of the revised paper.
>
> The evaluation framework and proposed metrics are generally applicable for all CL methods. However, the goal of our work is not to establish a comprehensive survey of existing methods, i.e. we focus on several representative CL methods rather than the many proposed variations. Nonetheless, in the revised paper we now discuss the mentioned works [a-c] in Appendix D.1.
> Additionally, in our new results for ER in online CL, we find that our metrics are successful in identifying the stability gap in Appendix D.1 (Table 5), where we can also clearly observe the stability gap in Figure 6. We refer to Appendix D.1 for a detailed discussion.
>
> **Q3: “The extreme case for continual learning is where the data examples are available only once, unlike the replay-based approaches, where the old data may be available for future training [d]. It would be interesting to apply the proposed framework to this challenging continual setup [d]”**
>
>
> We thank the reviewer for the useful reference [d]. However, it is not clear to us whether the reviewer refers to *‘online continual learning’* with *"data examples are available only once"*, or refers to *‘other methods than replay-based approaches’*.
> In case of online continual learning, we refer to our above answer on Q2. For methods besides replay-based methods, we elaborate in the following.
>
> Our study focuses on replay, knowledge-distillation, and model prior based methods that are commonly used in CL.
> For this rebuttal, we provide additional results on non-replay methods EWC and SI (model-prior based methods) and LwF (distillation-based method). In our significantly revised Section 5.1 we conduct experiments for EWC, SI and LwF both on the task-incremental and domain-incremental setting, where these methods can still be competitive w.r.t. replay [2]. We consistently observe the stability gap in both task-incremental Split-MNIST and domain-incremental Rotated-MNIST. For details we refer to the updated paper version with changes in blue.
>
> Parameter isolation and expansion methods (such as reference [d]) are often constrained to use the task-id during test-time, which can subsequently be used to activate only fixed parts of the network [3,4,5]. For example, PackNet [3] always has zero forgetting, as parts of the network are allocated to a task, and can not be changed while learning subsequent tasks. For clarity, we add a discussion on parameter-isolation methods in Appendix A, including the suggested reference [d].

---

> > ### Author Response · Authors · 2022-11-18
> > **Answer 2/2: Mentioned References**
> >
> > Please find below the references used in Answer 1/2.
> >
> > **References**:
> >
> > [1] Lopez-Paz, David, and Marc'Aurelio Ranzato. "Gradient episodic memory for continual learning." Advances in neural information processing systems 30 (2017).
> >
> > [2] Van de Ven, Gido M., and Andreas S. Tolias. "Three scenarios for continual learning." arXiv preprint arXiv:1904.07734 (2019).
> >
> > [3] Mallya, Arun, and Svetlana Lazebnik. "Packnet: Adding multiple tasks to a single network by iterative pruning." Proceedings of the IEEE conference on Computer Vision and Pattern Recognition. 2018.
> >
> > [4] Serra, Joan, et al. "Overcoming catastrophic forgetting with hard attention to the task." International Conference on Machine Learning. PMLR, 2018.
> >
> > [5] Rusu, Andrei A., et al. "Progressive neural networks." arXiv preprint arXiv:1606.04671 (2016).

---

> > ### Comment · Reviewer_q7YA · 2022-12-11
> > **Thanks for responce**
> >
> > Thank you for your clarification. I found this paper interesting and leaning toward acceptance. After observing other reviews and authors' replies to their issues, I recommend it for acceptance.

---

### Author Response · Authors · 2022-11-18
**Summary of the revised paper: New results and text improvements**

We thank the four reviewers for their valuable feedback. We hope to have addressed all raised concerns with extensive additional empirical evidence, and rephrasing parts of the paper for clarity.
We summarize our changes in the following. A revised version of the paper was submitted with all changes indicated in blue.

*Additional experiments:*
- Besides existing results for GEM, ER, and LwF, we additionally predict in our conceptual analysis and demonstrate in empirical experiments that the **stability gap also affects regularization-based methods EWC and SI** (Section 5.1).
- We go beyond the vision domain and show that the stability gap can also be observed with **speech classification** (Appendix D.8).
- We show for **ER** that our findings persist **over multiple memory sizes**, and find that even when all samples are stored (i.e., approximating sequential ‘joint’ training) the stability gap is still observed (Appendix D.2).
- Besides class and domain incremental learning, we extend the scope of our experiments to **task-incremental learning**. This is also referred to as the multi-head setting, where each task is allocated a separate classifier head (Section 5.1).
- Besides learning for multiple epochs per task, we now also consider **online Continual Learning** where samples are seen in a streaming fashion for only 1 epoch per task (Appendix D.1).

*Changes in text and notation for clarity:*
- Throughout the paper we made **notational changes to improve clarity**, and renamed our class-incremental benchmarks (MNIST, CIFAR10, Mini-Imagenet) to Split-MNIST, Split-CIFAR10, Split-MiniImagenet.
- We **reformulated the conceptual analysis of LwF** in Section 5.1 to improve clarity.
- In Appendix A, we **discuss the parameter isolation methods** in continual learning and how fixing parts of the model by design cannot result in a stability gap.

---

### Decision · Program_Chairs · 2023-01-20

**Decision:**

Accept: notable-top-25%

**Justification For Why Not Higher Score:**

This is a great analysis paper and is definitely worth a spotlight. However, I do not think that the stability gap is such a surprising phenomenon that deserves an oral. The paper formalizes some of the intuitions that are already known by the community in an informal way.

**Justification For Why Not Lower Score:**

I think the paper is interesting enough to be highlighted and hence my recommendation as a spotlight. This will motivate the design of new CL algorithms that aim to close the stability gap.

**Metareview: Summary, Strengths And Weaknesses:**

This paper makes an interesting observation about what happens when a continual learning system starts to train on a new task. The authors observe that the performance in the previous task drops and then recovers and call this the stability gap. The analysis shows that this stability gap happens in a wide variety of CL methods and they also propose continual evaluation metrics to capture this phenomenon. Overall this is a very interesting paper for the CL community and deserves to be published.

Reviewers raised some concerns about the conclusions being specific to image data and ER methods. But authors added speech data and also added more non-replay methods to show that this is a general phenomenon. I recommend accepting.

**Note From Pc:**

if the above contains the word "oral" or "spotlight" please see: "oral" presentation means -> notable-top-5% and "spotlight" means -> notable-top-25%. As stated in our emails, we are disassociating presentation type from AC recommendations